# COMBINING DIFFERENTIAL PRIVACY AND BYZANTINE RESILIENCE IN DISTRIBUTED SGD

## ABSTRACT

Privacy and Byzantine resilience (BR) are two crucial requirements of modern-day distributed machine learning. The two concepts have been extensively studied individually but the question of how to combine them effectively remains unanswered. This paper contributes to addressing this question by studying the extent to which the distributed SGD algorithm, in the standard parameter-server architecture, can learn an accurate model despite (a) a fraction of the workers being malicious (Byzantine), and (b) the other fraction, whilst being honest, providing noisy information to the server to ensure differential privacy (DP). We first observe that the integration of standard practices in DP and BR is not straightforward. In fact, we show that many existing results on the convergence of distributed SGD under Byzantine faults, especially those relying on $(\alpha, f)$-*Byzantine resilience*, are rendered invalid when honest workers enforce DP. To circumvent this shortcoming, we revisit the theory of $(\alpha, f)$-BR to obtain an approximate convergence guarantee. Our analysis provides key insights on how to improve this guarantee through hyperparameter optimization. Essentially, our theoretical and empirical results show that (1) an imprudent combination of standard approaches to DP and BR might be fruitless, but (2) by carefully re-tuning the learning algorithm, we can obtain reasonable learning accuracy while simultaneously guaranteeing DP and BR.

## 1 INTRODUCTION

Distributed machine learning (ML) has received significant attention in recent years due to the growing complexity of ML models and the increasing computational resources required to train them (Dean et al., 2012; Srivastava et al., 2015). One of the most popular distributed ML settings is the *parameter server* architecture, wherein multiple machines (called *workers*) jointly learn a single large model on their collective dataset with the help of a trusted *server* running the *stochastic gradient descent* (SGD) algorithm (Bottou, 2010). In this scheme, the server maintains an estimate of the model parameters, which is iteratively updated using stochastic gradients computed by the workers.

Compared to its centralized counterpart, distributed SGD is more susceptible to security threats. One of them is related to the violation of data privacy by an *honest-but-curious* server (Zhu et al., 2019). Another one is the malfunctioning due to (what is called) *Byzantine* behavior of workers (Lamport et al., 1982; Blanchard et al., 2017). In the past, significant progress has been made in addressing these issues separately. In the former case, $(\epsilon, \delta)$-*differential privacy* (DP) has become a dominant standard for preserving privacy in ML, especially when considering neural networks (Dwork et al., 2014; Abadi et al., 2016). In the latter case, $(\alpha, f)$-*Byzantine resilience* has emerged as the principal notion for demonstrating the Byzantine resilience (BR) of distributed SGD (Blanchard et al., 2017; El Mhamdi et al., 2018). Since DP and BR are two crucial pillars of distributed machine learning, practitioners will inevitably have to build systems satisfying both these requirements. It is thus natural to ask the following question: ***Can we simultaneously ensure DP and BR in distributed ML?***

In this paper, we take a first step towards a positive answer to this question by studying the resilience of the renowned DP-SGD algorithm (Abadi et al., 2016) against Byzantine workers. More precisely, we consider distributed SGD where, in each learning step, the honest workers inject Gaussian noise to their gradients to ensure $(\epsilon, \delta)$-DP, while the server updates the parameters by applying an $(\alpha, f)$-BR aggregation rule on the received gradients (to protect against Byzantine workers). Upon analyzing

the convergence of this algorithm, we show that DP and BR can indeed be combined, however, doing so is non-trivial. Our key contributions are summarized below.

**1. Inapplicability of existing results from the BR literature.** We start by highlighting an inherent incompatibility between the supporting theory of $(\alpha, f)$-BR and the Gaussian mechanism used in DP-SGD. Specifically, we show (in Section 3.2) that the *variance-to-norm* (VN) condition, critical to guarantee $(\alpha, f)$-BR, cannot be satisfied when honest workers enforce $(\epsilon, \delta)$-DP via Gaussian noise injection. Hence, existing results on the resilience of distributed SGD to Byzantine workers are not applicable when considering DP-SGD. More generally, this highlights limitations of many existing Byzantine resilient techniques in settings where the stochasticity of the gradients is non-trivial.

**2. Adapting the theory of BR to account for DP.** To overcome the aforementioned shortcoming, we introduce a relaxation of the VN condition (in Section 3.3), namely the *$\eta$-approximated* VN condition. By doing so, we (1) generalize existing results from the BR literature and (2) demonstrate *approximate* convergence of DP-SGD under Byzantine faults. Our convergence result can be roughly put as follows.

**Theorem** (Informal). *Let $Q$ be the loss function of the learning model, and $\theta_t$ be the parameter vector obtained after $t$ steps of our algorithm. If the $\eta$-approximated VN condition holds true, then*

$$\min_{t \in [T]} \mathbb{E}\left[\|\nabla Q(\theta_t)\|^2\right] \leq \max\left\{\eta^2, O\left(\frac{\log T}{\sqrt{T}}\right)\right\}$$

*where $[T] = \{1, \ldots, T\}$, and $\|\cdot\|$ denotes the Euclidean norm.*

As the aforementioned result suggests, a smaller $\eta$ ensures better convergence. To quantify this convergence guarantee, we present (in Section 3.4) necessary and sufficient conditions for the $\eta$-approximated VN condition to hold. Specifically, we show that the condition holds only if $\eta^2 \in \Omega\left(d \ln(1/\delta)/bm\epsilon\right)$ where $d$, $b$, and $m$ denote the model size, batch size, and dataset size respectively. This showcases an important interplay between DP and BR, e.g., larger $\epsilon$ and $\delta$ leads to stronger resilience to Byzantine workers at the expense of weaker privacy.

**3. From theoretical insights to practical convergence.**

Importantly, our result (in Section 3.4) provides key insights on how to better integrate standard approaches to DP and BR using *hyperparameter optimization* (HPO), e.g., by increasing the batch size $b$, or choosing an appropriate aggregation rule. The improvement is illustrated by a snippet of our experimental results in Figure 1. This finding is particularly interesting as these parameters have very little impact in most settings when considering DP or BR separately. We validate our theoretical insights in Section 4 through an exhaustive set of experiments on MNIST and Fashion-MNIST using neural networks.

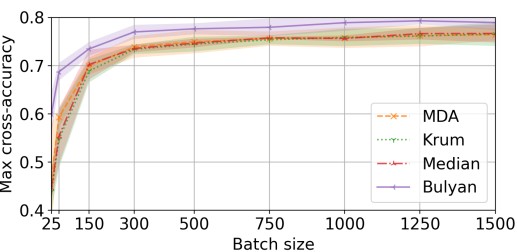

Figure 1: Impact of the batch size and aggregation rule on the cross-accuracy of DP-SGD against the *little* (Baruch et al., 2019) attack on Fashion-MNIST.

CLOSELY RELATED PRIOR WORKS

There has been a long line of research on the interplay between DP and other notions of robustness in ML (Dwork & Lei, 2009; Ma et al., 2019; Song et al., 2019b;a; Sun et al., 2019; Lécuyer et al., 2019; Pinot et al., 2019). However, previous approaches do not apply to our setting for two main reasons; (1) they do not address the privacy of the dataset against an honest-but-curious server, and (2) their underlying notion of robustness are either weaker than or orthogonal to BR. Furthermore, recent works on the combination of privacy and BR in distributed learning either study a weaker privacy model than DP or provide only elementary analyses (Chen et al., 2018; He et al., 2020; So et al., 2020; Guerraoui et al., 2021). We refer the interested reader to Appendix A for an in depth discussion of prior works. In short, we believe the present paper to be the first to provide an in-depth analysis with practical relevance on the integration of DP and BR in distributed learning.

## 2 PROBLEM SETTING AND BACKGROUND

Let $\mathcal{X}$ be the space of data points. We consider the parameter server architecture with $n$ workers $\{w_1, \ldots, w_n\}$ owning a common dataset $D \in \mathcal{X}^m$ of $m$ points. The workers seek to collaboratively compute a parameter vector $\theta \in \mathbb{R}^d$ that minimizes the empirical loss function $Q$ defined as follows:

$$Q(\theta) = \frac{1}{m} \sum_{x \in D} q(\theta, x) \quad \forall \theta \in \mathbb{R}^d, \tag{1}$$

where $q$ is a point-wise *loss function*. We assume that function $Q$ is differentiable and admits a non-trivial local minimum. In other words, $\nabla Q$ admits a critical point, but it is not null everywhere. We also make the following standard assumptions.

**Assumption 1** (Bounded norm). *There exists a finite real $C < \infty$ such that for all $\theta \in \mathbb{R}^d$ and $x \in D$,*

$$\|\nabla q(\theta, x)\| \leq C.$$

**Assumption 2** (Bounded variance). *There exists a real value $\upsilon < \infty$ such that for all $\theta \in \mathbb{R}^d$,*

$$\frac{1}{m} \sum_{x \in D} \|\nabla q(\theta, x) - \nabla Q(\theta)\|^2 \leq \upsilon^2.$$

**Assumption 3** (Smoothness). *There exists a real value $L < \infty$ such that for all $\theta, \theta' \in \mathbb{R}^d$,*

$$\|\nabla Q(\theta) - \nabla Q(\theta')\| \leq L \|\theta - \theta'\|.$$

Assumptions 2 and 3 are classical to most optimization problems in machine learning (Bottou et al., 2018). Assumption 1 is merely used to avoid unnecessary technicalities, especially when studying differential privacy. In practice, it can be easily enforced by gradient clipping (Abadi et al., 2016).

In an ideal setting, when all the workers are honest (i.e., non-Byzantine) and data privacy is not an issue, a standard approach to solving the above learning problem is the distributed implementation of the stochastic gradient descent (SGD) method. In this algorithm, the server maintains an estimate of the parameter vector which is updated iteratively by using the *average* of the gradient estimates sent by the workers. However, this algorithm is vulnerable to both privacy and security threats.

**Threat model.** We consider the server to be *honest-but-curious*, and that some of the workers are *Byzantine*. An honest-but-curious server follows the prescribed algorithm correctly, but may infer sensitive information about workers' data using their gradients and any other additional information that can be gathered during the learning as demonstrated by Zhu et al. (2019). On the other hand, Byzantine workers need not follow the prescribed algorithm correctly and can send arbitrary gradients. For instance, they may either *crash* or even send adversarial gradients to prevent convergence of the algorithm (Blanchard et al., 2017).

### 2.1 DISTRIBUTED SGD WITH DIFFERENTIALLY PRIVACY

Over the last decade, differential privacy (DP) has become a gold standard in privacy-preserving data analysis (Dwork et al., 2014). Intuitively, a randomized algorithm is said to preserve DP if its executions on two adjacent datasets are indistinguishable. More formally, two datasets $D$ and $D'$ are said to be adjacent if they differ by at most one sample. Then, $(\epsilon, \delta)$-DP is defined as follows.

**Definition 1** ($(\epsilon, \delta)$-DP). *Let $\epsilon > 0$, $\delta \in [0, 1]$ and $\mathcal{O}$ an arbitrary output space. A randomized algorithm $\mathcal{M} : \mathcal{X}^m \to \mathcal{O}$ is $(\epsilon, \delta)$-differentially private if for any two adjacent datasets $D, D'$, and any possible set of outputs $O \subset \mathcal{O}$,*

$$\mathbb{P}[\mathcal{M}(D) \in O] \leq e^\epsilon \mathbb{P}[\mathcal{M}(D') \in O] + \delta.$$

By far, the most widely used approach to ensure DP in machine learning is to use the differentially private version of SGD, called DP-SGD (Song et al., 2013; Bassily et al., 2014; Abadi et al., 2016). The distributed implementation of this scheme against an honest-but-curious server consists, at every step, in making the honest workers add Gaussian noise with variance $s^2$ to their stochastic gradients before sending them to the server. When $s$ is chosen appropriately (e.g., see Theorem 1), each learning step satisfies $(\epsilon, \delta)$-DP at the worker level. Finally, the privacy guarantee of the overall learning procedure is obtained by using the *composition property* of DP (Kairouz et al., 2015; Abadi et al., 2016; Wang et al., 2019). However, we are mainly interested in studying the impact of *per-step and per-worker* privacy budget $(\epsilon, \delta)$ on the resilience of the algorithm to Byzantine workers.

## 2.2 BYZANTINE RESILIENCE OF DISTRIBUTED SGD

In the presence of Byzantine workers, the server can no longer rely on the average of workers' gradients to update the model parameters. Instead, it uses a gradient aggregation rule (GAR) $F : \mathbb{R}^{d \times n} \to \mathbb{R}^d$ that is resilient to incorrect gradients that may be sent by at most $f$ Byzantine workers. A standard notion for defining this resilience is $(\alpha, f)$-Byzantine resilience stated below, which was originally proposed by Blanchard et al. (2017).

**Definition 2** ($(\alpha, f)$-*Byzantine resilience*)**.** *Let* $0 \leq \alpha < \pi/2$, *and* $0 \leq f < n$. *Consider* $n$ *random vectors* $g^{(1)}, \ldots, g^{(n)}$ *among which at least* $n - f$ *are i.i.d. from a common distribution* $\mathcal{G}$. *Let* $G \sim \mathcal{G}$ *be a random vector characterizing this distribution. A GAR* $F$ *is said to be* $(\alpha, f)$-*Byzantine resilient for* $\mathcal{G}$ *if its output* $R = F(g^{(1)}, \ldots, g^{(n)})$ *satisfies the following two properties:*

1. $\langle \mathbb{E}[R], \mathbb{E}[G] \rangle \geq (1 - \sin \alpha) \|\mathbb{E}[G]\|^2 > 0$, *and*

2. *for any* $r \in \{2, 3, 4\}$, $\mathbb{E}[\|R\|^r]$ *is upper bounded by a linear combination of* $\mathbb{E}[\|G\|^{r_1}], \ldots, \mathbb{E}[\|G\|^{r_k}]$ *where* $\sum_{1 \leq i \leq k} r_i = r$.

This condition has been shown critical to ensure convergence of the distributed SGD algorithm in the presence of up to $f$ Byzantine workers (Blanchard et al., 2017; El Mhamdi et al., 2018). Thus, it serves as an excellent starting point for studying the Byzantine resilience of distributed DP-SGD. Consequently, we consider the algorithm where the server implements a Byzantine robust GAR while the honest workers follow instructions prescribed in DP-SGD.

## 3 COMBINING DIFFERENTIAL PRIVACY AND BYZANTINE RESILIENCE

Algorithm 1, described below, combines the standard techniques to DP and BR in distributed SGD. Given a GAR $F$ and a noise injection parameter $s$, Algorithm 1 computes $T$ steps of distributed DP-SGD with $F$ as an aggregation rule at the server to guarantee BR.

---

**Algorithm 1:** Distributed DP-SGD with Byzantine resilience

---

**Setup:** The server chooses an arbitrary initial parameter vector $\theta_1 \in \mathbb{R}^d$, learning rates $\{\gamma_1, \ldots, \gamma_T\}$, and a deterministic GAR $F$. Honest workers have a fixed batch size $b$ and noise injection parameter $s$.

**for** $t = 1, \ldots, T$ **do**

    The server broadcasts $\theta_t$ to all workers.

    **foreach** honest worker $w_i$ **do**

        1. $w_i$ builds a set $B_t$ by sampling $b$ points at random without replacement from $D$ and computes a noisy gradient estimate with noise injection parameter $s$, i.e., it computes

$$g_t^{(i)} = \frac{1}{|B_t|} \sum_{x^{(i)} \in B_t} \nabla q\left(\theta_t, x^{(i)}\right) + y_t^{(i)}; \quad y_t^{(i)} \sim \mathcal{N}(0, s^2 I_d). \tag{2}$$

        2. $w_i$ sends the resulting noisy gradient to the server.

    **end**

    **foreach** Byzantine worker $w_i$ **do**

        $w_i$ sends to the server a (possibly arbitrary) vector $g_t^{(i)}$ as its "gradient".

    **end**

    The server computes the aggregate $R_t$ of the received gradients using $F$, i.e., it computes

$$R_t = F\left(g_t^{(1)}, \ldots, g_t^{(n)}\right). \tag{3}$$

    The server updates the parameter vector using the learning rate $\gamma_t$ as follows

$$\theta_{t+1} = \theta_t - \gamma_t R_t. \tag{4}$$

**end**

---

Note that when $s = 0$, i.e., when no noise is injected, Algorithm 1 reduces to a classical Byzantine resilient distributed SGD algorithm as presented in prior works such as Blanchard et al. (2017) and El Mhamdi et al. (2018). Furthermore, when $f = 0$ and $F$ is the average function, it reduces to a distributed implementation of the well-known DP-SGD scheme, e.g., from Abadi et al. (2016).

## 3.1 DIFFERENTIAL PRIVACY GUARANTEE

Intuitively, Algorithm 1 should inherit the privacy guarantees of DP-SGD. Indeed, the privacy preserving scheme applied at the worker level is the same and will not by altered by the GAR thanks to the post-processing property of DP (Dwork et al., 2014). Then, owing to previous works, we can easily show that Algorithm 1 satisfies $(\epsilon, \delta)$-DP at each step and for each honest worker when $s \approx {2C}/{b\epsilon}\sqrt{2\ln\left({1.25}/{\delta}\right)}$. Furthermore, as shown in Theorem 1, we can obtain a much tighter analysis using advanced analytical tools such as *privacy amplification via sub-sampling* (Balle et al., 2018).

**Theorem 1.** *Suppose that Assumption 1 holds true. Let $b \in [m]$ and $(\epsilon, \delta) \in (0, 1)^2$. Consider Algorithm 1 with $s = \frac{2C}{b\ln\left((e^\epsilon - 1)\frac{m}{b} + 1\right)}\sqrt{2\ln\left(\frac{1.25b}{m\delta}\right)}$. Then, each honest worker satisfies $(\epsilon, \delta)$-DP at every step $t \in [T]$ of the procedure.*

Henceforth, whenever we refer to Algorithm 1 with per-step and per-worker privacy budget $(\epsilon, \delta)$, we will consider $s$ as defined in Theorem 1 above.

## 3.2 INAPPLICABILITY OF EXISTING RESULTS FROM THE BR LITERATURE

As discussed in Section 2, prior works on BR can demonstrate the convergence of Algorithm 1 *if* the GAR $F$ is $(\alpha, f)$-Byzantine resilient during the entire learning process. However, verifying the validity of $(\alpha, f)$-BR is nearly impossible as the condition depends upon the gradients of the Byzantine workers that can be arbitrary (Blanchard et al., 2017). The only verifiable condition known in the literature to guarantee $(\alpha, f)$-BR is the *variance-to-norm* (VN) condition, which is defined as follows (El Mhamdi et al., 2018).

**Definition 3** (VN Condition). *For a parameter vector $\theta \in \mathbb{R}^d$, let $G(\theta)$ denote the random vector characterizing the gradients sent by the honest workers to the server at $\theta$. A GAR $F$ satisfies the VN condition if for any $\theta$ such that $G(\theta)$ has a non-zero mean,*

$$\kappa_F(n, f)^2 \, \mathbb{E}\left[\|G(\theta) - \mathbb{E}\left[G(\theta)\right]\|^2\right] < \|\mathbb{E}\left[G(\theta)\right]\|^2$$

*where $\kappa_F(n, f) > 0$ is the* multiplicative constant *of GAR $F$ that depends on $n$ and $f$.*[1]

This condition means that for a GAR $F$ to guarantee convergence for the procedure, the distribution of the gradient estimates at parameter $\theta$ must be "well-behaved". For instance, if the norm of the expected stochastic gradients converges to $0$ then so should the variance. Note that in the case of Algorithm 1, from (2) we obtain that for any $\theta \in \mathbb{R}^d$,

$$G(\theta) := \frac{1}{b}\sum_{x \in B}\nabla q\left(\theta, x\right) + y \tag{5}$$

where $B$ is a set of $b$ data points sampled randomly without replacement from $D$, and $y \sim \mathcal{N}(0, s^2 I_d)$. Thus, the VN condition can no longer be satisfied whenever $s > 0$, i.e., workers follow instructions prescribed in DP-SGD. We show this formally in Proposition 1 below.

**Proposition 1.** *Let $b \in [m]$. Consider Algorithm 1 with $s > 0$. If Assumption 3 holds true, then there exists no GAR that satisfies the VN condition.*

Note that when $\epsilon$ and $\delta$ are non-zero, we will have $s > 0$ as explained in Section 3.1. Accordingly, Proposition 1 means that prior results on the convergence of existing Byzantine resilient GARs, including the works by Blanchard et al. (2017) and El Mhamdi et al. (2018), are no longer valid when enforcing any *non-zero* level of DP. Although the VN condition is only a sufficient one, due to the lack of necessary conditions in the literature, it is the most widely used tools for proving BR, e.g., see Blanchard et al. (2017); El Mhamdi et al. (2018); Xie et al. (2018b); El-Mhamdi et al. (2020); Boussetta et al. (2021). Hence, Proposition 1 highlights an inherent limitation of the theory of BR, especially when simultaneously enforcing DP via noise injection.

---

[1]Precise values of $\kappa_F(n, f)$ for most popular GARs can be found in Appendix B.

### 3.3 Adapting the Theory of BR to Account for DP

To circumvent the aforementioned limitation, we propose a relaxation of the theory of $(\alpha, f)$-BR by relaxing the original VN condition to the $\eta$-*approximated* VN condition defined below.

**Definition 4** ($\eta$-approximated VN condition). *Let $G(\theta)$ denote the random vector characterizing the gradients sent by the honest workers to the server at parameter vector $\theta$. For $\eta \geq 0$, a GAR $F$ satisfies the $\eta$-approximated VN condition if for all $\theta \in \mathbb{R}^d$ such that $\|\mathbb{E}[G(\theta)]\| > \eta$,*

$$\kappa_F(n, f)^2 \, \mathbb{E}\left[\|G(\theta) - \mathbb{E}[G(\theta)]\|^2\right] < \|\mathbb{E}[G(\theta)]\|^2$$

*where $\kappa_F(n, f) > 0$ is the multiplicative constant of GAR $F$ that depends on $n$ and $f$.*

Definition 4 relaxes the initial VN condition by allowing a subset of (possible) parameter vectors $\theta$ to violate the inequality in Definition 3. In particular, as $\mathbb{E}[G(\theta)] = \nabla Q(\theta)$, when the gradients are sufficiently close to a local minimum, or $\|\nabla Q(\theta)\| \leq \eta$, the inequality need not be satisfied. While the $\eta$-approximated VN condition is a natural extension of Definition 3, it enables us to study cases where the distribution of the gradients at $\theta$ is non-trivial, e.g., the variance of $G(\theta)$ need not vanish when $\nabla Q(\theta)$ approaches 0. Consequently, we can utilize this new criterion to analyze the convergence of Algorithm 1 for different GARs and levels of privacy. Assuming $\eta$-approximated VN condition, we show in Theorem 2 the approximate convergence of Algorithm 1.

**Theorem 2.** *Let $\eta \geq 0$ and $b \in [m]$. Consider Algorithm 1 with $s > 0$, a GAR $F$ satisfying the $\eta$-approximated VN condition, and $\gamma_t = 1/\sqrt{t}$ for all $t \in [T]$. If Assumptions 1, 2, and 3 hold true, then there exists $\alpha \in [0, \pi/2]$ and $\mu \in [0, \infty)$ such that for any $T \geq 1$,*

$$\min_{t \in [T]} \mathbb{E}\left[\|\nabla Q(\theta_t)\|^2\right] \leq \max\left\{\eta^2, \frac{Q(\theta_1) - Q^*}{(1 - \sin \alpha)}\left(\frac{1}{\sqrt{T}}\right) + \frac{\mu \sigma^2 L}{2(1 - \sin \alpha)}\left(\frac{1 + \ln T}{\sqrt{T}}\right)\right\}$$

*where $Q^*$ is the minimum value of Q, i.e., $Q^* = \min_{\theta \in \mathbb{R}^d} Q(\theta)$, and $\sigma = \sqrt{v^2 + ds^2 + C^2}$.*

According to Theorem 2, Algorithm 1 can compute a parameter $\theta$ for which $\|\nabla Q(\theta)\| \leq \eta$ in expectation with a rate of $O(\ln T/\sqrt{T})$. In other words, when the loss function $Q$ is *regularized* (see, e.g., Bottou et al. (2018)), it finds an approximate local minimum with an error proportional to $\eta^2$. Note that, when $\eta = 0$ (i.e., when DP is not enforced), the above result encapsulates the existing convergence results from the BR literature, e.g., Blanchard et al. (2017); El Mhamdi et al. (2018).

**Remark 1.** *For generality, we do not provide the exact values for parameters $\alpha$ and $\mu$ in Theorem 2. These two constants depend on the learning scheme that is applied, in particular the resilience properties of the GAR used. However, since these parameters are constant throughout the learning procedure, keeping them to be generic does not affect our conclusions on the asymptotic error.*

### 3.4 Studying the Interplay between DP and BR

The value of $\eta$ is intrinsically linked to the amount of noise that workers inject to the procedure. In a way, it represents the impact of per-worker DP on the resilience of Algorithm 1 to Byzantine workers. To quantify this impact, we present in Proposition 2 sufficient and necessary conditions for a GAR $F$ to satisfy the $\eta$-approximated VN condition in the context of Algorithm 1.

**Proposition 2.** *Let $b \in [m]$, $(\epsilon, \delta) \in (0, 1)^2$. Consider Algorithm 1 with privacy budget $(\epsilon, \delta)$ and GAR $F$ with multiplicative constant $\kappa_F(n, f) > 0$. Then, the following assertions hold true.*

1. *Under Assumptions 1 and 3, the $\eta$-approximated VN condition can hold true only if*

$$\eta^2 \geq 4\kappa_F(n, f)^2 C^2 d \ln\left(\frac{1.25b}{m\delta}\right) \frac{1}{bm(e^\epsilon - 1)}.$$

2. *Additionally under Assumption 2, if*

$$\eta^2 \geq \kappa_F(n, f)^2 \left(8C^2 d \ln\left(\frac{1.25b}{m\delta}\right)\left(\frac{1}{m(e^\epsilon - 1)} + \frac{1}{b}\right)^2 + v^2\right)$$

*then $F$ satisfies the $\eta$-approximated VN condition.*

The above result, in conjunction with Theorem 2, presents a convergence guarantee that can be obtained by distributed DP-SGD under Byzantine faults. In particular, we have the following corollary of Theorem 2 and Proposition 2.

**Corollary 1.** *Let $b \in [m]$, $(\epsilon, \delta) \in (0, 1)^2$. Consider Algorithm 1 with privacy budget $(\epsilon, \delta)$ and GAR F, and $\gamma_t = 1/\sqrt{t}$ for all $t \in [T]$. If Assumptions 1, 2, and 3 are satisfied, then for any $T \geq 1$,*

$$\min_{t \in [T]} \mathbb{E}\left[\|\nabla Q(\theta_t)\|^2\right] \leq \max\left\{\kappa_F(n,f)^2\left(8C^2 d \ln\left(\frac{1.25b}{m\delta}\right)\left(\frac{1}{m(e^\epsilon - 1)} + \frac{1}{b}\right)^2 + \upsilon^2\right), O\left(\frac{\ln T}{\sqrt{T}}\right)\right\}.$$

Corollary 1 quantifies the impact of different parameters on the convergence of the algorithm. For instance, we observe that larger values of $\epsilon$ and $\delta$, i.e., weaker DP guarantees, imply smaller worst-case convergence error and therefore, better guarantee of learning. But importantly, it also shows how the convergence guarantee of the algorithm depends upon other hyperparameters, namely the batch size $b$, the number of parameters $d$, and the multiplicative constant $\kappa_F(n, f)$ of the GAR. Let us for example take the case of the batch size below.

**Impact of batch size.** We consider the specific GAR of Minimum-Diameter Averaging (*MDA*) for which $\kappa_{MDA}(n, f) = \sqrt{8}f/n-f$ (El Mhamdi et al., 2018). Then, from Corollary 1, we obtain that

$$\min_{t \in [T]} \mathbb{E}\left[\|\nabla Q(\theta_t)\|^2\right] \leq \max\left\{\frac{8f^2}{(n-f)^2}\left(8C^2 d \ln\left(\frac{1.25b}{m\delta}\right)\left(\frac{1}{m(e^\epsilon - 1)} + \frac{1}{b}\right)^2 + \upsilon^2\right), O\left(\frac{\ln T}{\sqrt{T}}\right)\right\}.$$

From above, we note that when parameters $\epsilon$ and $\delta$ are in the interval $(0, 1)$ and $f > 0$, i.e., both DP and BR are enforced, then increasing the batch size $b$ indeed reduces the asymptotic convergence error of the algorithm. However, this is not the case when we consider DP and BR separately. When all workers are honest, $f = 0$, which implies $\eta = 0$. The algorithm then asymptotically converges to a local minimum regardless of the batch size used. On the other hand, when the workers do not obfuscate their gradients ($s = 0$), the $\eta$-approximated VN condition holds true for $\eta \geq \left(\sqrt{8}f/n-f\right)\upsilon$. Then, the asymptotic convergence error of the algorithm is again independent of the batch size. To conclude, the batch size plays a crucial role in improving the learning accuracy when enforcing DP and BR simultaneously, but it should have little influence when considering them individually.

**Remark 2.** *Although Corollary 1 provides some useful insights on improving the accuracy of the learning algorithm combining DP and BR, it need not be* tight *as it only provides an upper bound relying on a sufficient condition; the $\eta$-approximated VN condition. It turns out that providing a non-trivial lower bound for distributed SGD in the presence of Byzantine faults remains an open problem, even without DP. In spite of this, we show the practical relevance of the insights obtained from Corollary 1 through an exhaustive set of experiments in the subsequent section.*

## 4 NUMERICAL EXPERIMENTS

The goal of our experiments is to investigate whether our theoretical insights are actually applicable in practice and whether hyperparameter optimization (HPO) can improve the integration of DP and BR. Accordingly, we assess the impact of varying different hyperparameters on the training losses and top-1 cross-accuracies of a neural network under $(\epsilon, \delta)$-DP and attacks from Byzantine workers over a maximum of 300 learning steps.

**Reproducibility.** All our experiments (training + graphs) are reproducible in one command. Please see `code/README.md` in the supplementary material. Additional graphs are available in `plots/`.

### 4.1 EXPERIMENTAL SETUP

**Datasets.** We use MNIST (LeCun & Cortes, 2010) and Fashion-MNIST (Xiao et al., 2017). The datasets are pre-processed before training. MNIST receives an input image normalization with mean 0.1307 and standard deviation 0.3081. Fashion-MNIST is expanded with horizontally flipped images. Due to space limitations, we only showcase here results on the Fashion-MNIST dataset.

**Architecture and fixed hyperparmaters.** We consider a feed-forward neural network composed of two fully-connected linear layers of respectively 784 and 100 inputs (for a total of $d = 79\,510$ parameters) and terminated by a *softmax* layer of 10 dimensions. ReLU is used between the two

linear layers. We use the Cross Entropy loss, a total number of workers $n = 15$, Polyak's momentum of $0.99$ at the workers, a constant learning rate of $0.5$, and a clipping parameter $C = 2$. We also add an $\ell_2$-regularization factor of $10^{-4}$. Note that some of these constants are reused from the literature on BR, especially from Baruch et al. (2019); Xie et al. (2019); El-Mhamdi et al. (2021).

**Varying hyperparameters for HPO.** For both datasets, we vary the batch size $b$ within $\{25, 50, 150, 300, 500, 750, 1000, 1250, 1500\}$, the per-step and per-worker privacy parameter $\epsilon$ in $\{0.2, 0.1, 0.05\}$ ($\delta$ is fixed to $10^{-5}$), the number of Byzantine workers $f$ in $\{3, 6\}$ as well as the attack they implement (*little* from Baruch et al. (2019) and *empire* from Xie et al. (2019)). We also vary the Byzantine resilient GAR $F$ in $\{MDA, Krum, Median, Bulyan\}$. Note that due to its large computational cost, we only use the *Bulyan* aggregation rule when $f = 3$.

Each of the 432 possible combinations of these hyperparameters is run 5 times using seeds from 1 to 5 (for reproducibility purposes), totalling in 2160 runs. Each run satisfies $(\epsilon, \delta)$-DP at every step under attacks from Byzantine workers. To assess the impact of the privacy noise alone, we also run the experiments specified above with the *averaging* GAR and without Byzantine workers (denoted by "No attack"). These experiments account for another 27 combinations, totalling in 135 additional runs. Overall, we performed a comprehensive set of 2295 runs for which we provide a brief summary below. More details on the experimental setup and results can be found in Appendices D and E.

## 4.2 EXPERIMENTAL RESULTS

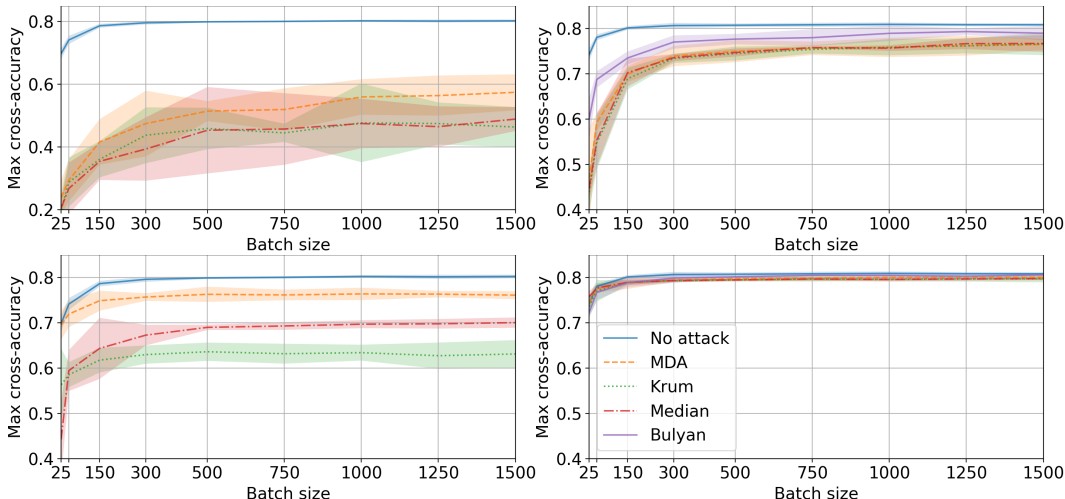

Figure 2: Maximum top-1 cross-accuracy reached on Fashion-MNIST when *only* varying the batch size $b$ for different threat scenarios and different GARs. The first and second rows show the *little* and *empire* attacks respectively. The first and second columns display $f = 6, \epsilon = 0.05$ and $f = 3, \epsilon = 0.2$ respectively. All reported metrics include a standard deviation obtained with the 5 consecutive runs. As a reference, note that the maximal top-1 cross-accuracy achieved by the tested model in the vanilla setting (i.e., with neither DP nor Byzantine faults) is around 84–85% for Fashion-MNIST.

In Figure 2, we give a snapshot of our results by showcasing 4 characteristic outcomes encountered. Below, we present further characterization of them. Besides validating our theoretical insights on the impact of the batch size and the GAR selection on the convergence of Algorithm 1, these plots also showcase the threat scenarios in which hyperparameter optimization (HPO) has the most impact. Note that the *little* attack was more damaging than *empire* in our experiments; hence in our discussion below, we consider *little* to be a stronger threat than *empire*, ceteris paribus.

1. *Strongest threat scenario (top left).* We consider *little* with $f = 6$ and $\epsilon = 0.05$, i.e., the strongest level of attack and privacy we implemented. In this stringent scenario, the algorithm fails to deliver good learning accuracy under Byzantine attacks. Although increasing the batch size helps improve the convergence, the accuracy remains quite poor (well below 0.6, even when $b = 1500$).

2. *Relaxed threat scenario (bottom left).* Here, we keep $f = 6$ and $\epsilon = 0.05$, but we trade the attack for a weaker one (*empire*). This scenario validates our intuition on the advantage of increasing the batch size, but it mostly highlights the impact of GAR selection. Different GARs differ significantly in their maximum cross accuracies, while *MDA* performs the best.

3. *Mild threat scenario (top right):* We now consider $\epsilon = 0.2$ and $f = 3$, i.e., a weaker privacy guarantee and a fewer number of Byzantine workers. However, we revert back to *little* attack. We see that, for all GARs, increasing the batch size significantly improves the maximum cross-accuracy. The choice of GAR also impacts the performance, with *Bulyan* being the best.

4. *Weakest threat scenario (bottom right):* We consider *empire* with $f = 3$ and $\epsilon = 0.2$. The threat is so weak that all GARs perform almost the same. Although HPO still helps to obtain a better accuracy, it is not critical in this setting.

**Main Takeaway.** Our empirical results show that training a feed-forward neural network under both DP and BR is *possible* but *expensive* in some settings. Indeed, in the non-trivial threat scenarios, to achieve the same maximum cross-accuracy as DP-SGD with $b \approx 50$, we need a per-worker batch size $\geq 1000$, i.e., 20 times larger than the Byzantine-free setting. Moreover, depending upon the setting, the selection of the GAR might be more influential than the batch size. Finally, note that in the Byzantine-free setting, the DP-SGD algorithm obtains reasonable cross-accuracies (close to $0.8$) for most batch sizes considered. This validates our theoretical findings (discussed in Section 3.4) that the batch size has a more significant impact when combining DP and BR compared to when enforcing DP alone. Similar observations on the negligible impact of the batch size in the privacy-free setting (but under Byzantine attacks) can be found in Appendix E.

## 5 Conclusion & Open problems

In this paper, we have studied the integration of standard approaches to DP and BR, namely the distributed implementation of the popular DP-SGD protocol in conjunction with $(\alpha, f)$-BR GARs. Upon highlighting the limitations of the existing theory of BR when applied to this algorithm, we have proposed a generalization of this theory. By doing so, we have (1) quantified the impact of DP on BR, and (2) proposed an HPO scheme to effectively combine DP and BR. Our results have shown that DP and BR can be combined but at the expense of computational cost in some settings.

Our generalization of the theory of $(\alpha, f)$-BR is also of independent interest. Specifically, we have proposed a relaxation of the VN condition as $\eta$-approximated VN condition. Although the VN condition is quite stringent and only sufficient, it is consistently relied upon to design and study different Byzantine resilient GARs (Blanchard et al., 2017; El Mhamdi et al., 2018; El-Mhamdi et al., 2020; 2021; Boussetta et al., 2021). Hence, our convergence result, obtained using the relaxed $\eta$-approximated VN condition, supersedes many existing results in the literature of BR.

Interestingly, we have observed through our experiments (see Appendix E.2) that even when the relaxed $\eta$-approximated VN condition is violated, the algorithm obtains reasonable learning accuracy. This observation opens two interesting problems expounded below.

1. *A theoretical problem:* The VN condition (either approximated or not) is not tight enough to fully characterize BR. That is, in some cases, a GAR may be $(\alpha, f)$-BR without satisfying the VN condition. Furthermore, the theory of BR focuses on "worst-case" attacks that, for now, might not be achievable in practice. Hence, the question on the tightness of the VN condition for any specific attack, even without DP, remains open.

2. *An empirical problem:* The practice of BR focuses on state-of-the-art realizable attacks. These attacks are arguably sub-optimal explaining why we can obtain reasonable learning accuracy despite the violation of the VN condition. This also calls for designing better (or stronger) attacks.

Finally, while we have focused on adapting the theory of BR to make it more compatible with the standard DP-SGD algorithm, an alternate future direction could be to investigate other DP mechanisms that may comply better with classical approaches to BR, while preserving DP guarantees.

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

# A   RELATED WORK

**Privacy.**  In the past, significant attention has been given to protecting data privacy for both centralized (Song et al., 2013; Damaskinos et al., 2021; Abadi et al., 2016) and distributed SGD (Shokri & Shmatikov, 2015; Naseri et al., 2020). Although several techniques for data protection exist such as the encryption (Tang et al., 2019) or obfuscation (Gade & Vaidya, 2018) of gradients, the most standard approach consists in adding DP noise to the gradients computed by the workers (Abadi et al., 2016; Song et al., 2013; Shokri & Shmatikov, 2015; Naseri et al., 2020), which is what we consider. However, these works only consider a fault-free setting where all workers are assumed to be honest.

**Byzantine resilience.**  In a separate line of research, several other works have designed Byzantine resilient schemes for distributed SGD in the parameter-server architecture (Blanchard et al., 2017; El Mhamdi et al., 2018; Yin et al., 2018; Xie et al., 2018a;b; El-Mhamdi et al., 2020; Boussetta et al., 2021). Nevertheless, in these papers, the training data is not protected, meaning that their methods do not consider the privacy threat associated with sharing unencrypted gradients with the server.

**Combining privacy and BR.**  Although scarce, there has been some work on tackling the problem of combining privacy and BR. For instance, He et al. (2020) consider this problem for a different framework that includes two honest-but-curious *non-colluding* servers, a strong assumption that does not always hold in practice. Furthermore, their *additive secret sharing* scheme is rendered ineffective in our setting where there is a single honest-but-curious server that obtains information from all the workers. In the context of privacy, the single-server setting generalizes the multi-server setting with *colluding* servers. Another related work, the *BREA* framework, proposes the use of *verifiable secret sharing* amongst workers (So et al., 2020). However, the presented privacy scheme scales more poorly than DP mechanisms, and is infeasible in most distributed ML settings with no inter-worker communication. Chen et al. (2018) propose the *LearningChain* framework that is claimed to combine DP and BR. However, *LearningChain* is an experimental method, and Chen et al. do not provide any formal guarantees either on the resilience or on the convergence of the proposed algorithm.

Recently, Guerraoui et al. (2021) studied the problem of satisfying both DP and BR in a single-server distributed SGD framework. While they demonstrate the computational hardness of this problem in practice, we go beyond by showing an inherent incompatibility between the supporting theory of $(\alpha, f)$-BR and the Gaussian mechanism from DP. Moreover, our approximate convergence result generalizes the prior works on BR. This generalization is critical to quantifying the interplay between DP and BR. Importantly, while Guerraoui et al. (2021) only give elementary analysis explaining the difficulty of the problem, we show that a careful analysis can help combine DP and BR.

**Studying the interplay between DP and other notions of robustness.**  There has been a long line of work studying the interplay and mutual benefits of DP and robustness to data corruption in the centralized learning setting (Dwork & Lei, 2009; Ma et al., 2019). However, these works do not consider the problem of a distributed scenario with an honest-but-curious server, and they are not applicable to our setting. Furthermore, data corruption is actually a weaker threat than BR as the adversary cannot select its gradients online to disrupt the learning process.

Recently, there have been some work on the interplay between DP and robustness to evasion attacks (a.k.a. adversarial examples). Interestingly, some findings in that line of research are similar to ours. DP and robustness to adversarial examples have been demonstrated to be very close from a high-level theoretical point of view even if their semantics are very different (Lécuyer et al., 2019; Pinot et al., 2019). However, some recent works have pointed out that these two notions might be conflicting in some settings (Song et al., 2019a;b). It is however worth noting that BR and robustness to adversarial examples are two orthogonal concepts. In particular, the robustness of a model (at testing time) to evasion attacks does not provide any guarantee on the robustness (at training time) to Byzantine behaviors. Similarly, as BR focuses on the training (optimization) procedure, we can always train models using a Byzantine resilient aggregation rule but without obtaining robustness to evasion attacks. The connection between these two notions of robustness remains an open problem.

Finally, Sun et al. (2019) suggested that in the context of federated learning, differential privacy could help defend against backdoor attacks. However, this hypothesis got challenged by Wang et al. (2020).

## B    STANDARD GARs WITH ASSOCIATED MULTIPLICATIVE CONSTANTS

In this section, we present the different GARs used in our experiments, along with their associated VN conditions (Definition 3) and multiplicative constants $\kappa_F(n, f)$.

### B.1    KRUM

*Krum* is an aggregation rule introduced under the assumption that $n \geq 2f + 3$. It consists in selecting the gradient which has the smallest mean squared distance, where the mean is computed over its $n - f - 2$ closest gradients (Blanchard et al., 2017). Formally, let $g^{(1)}, g^{(2)}, ..., g^{(n)}$ be the gradients received by the parameter server. For any $i \in \{1, 2, ..., n\}$ and $j \neq i$, we denote by $i \to j$ the fact that $g^{(j)}$ is amongst the $n - f - 2$ closest vectors (in distance) to $g^{(i)}$ within the submitted gradients. *Krum* assigns to each $g^{(i)}$ a score

$$s_i = \sum_{j : i \to j} \left\| g^{(i)} - g^{(j)} \right\|^2, \tag{6}$$

and outputs the gradient with the lowest score. Blanchard et al. (2017) prove that *Krum* is $(\alpha, f)$-Byzantine resilient, assuming that the following VN condition is satisfied:

$$2 \left( n - f + \frac{f(n - f - 2) + f^2(n - f - 1)}{n - 2f - 2} \right) \mathbb{E} \left[ \| G(\theta) - \mathbb{E}\left[ G(\theta) \right] \|^2 \right] < \| \mathbb{E}\left[ G(\theta) \right] \|^2. \tag{7}$$

Therefore, the multiplicative constant for *Krum* is

$$\kappa_{Krum}(n, f) = \sqrt{2 \left( n - f + \frac{f(n - f - 2) + f^2(n - f - 1)}{n - 2f - 2} \right)}. \tag{8}$$

### B.2    MINIMUM-DIAMETER AVERAGING (*MDA*)

*MDA* is an aggregation rule introduced under the assumption that $n \geq 2f + 1$. It outputs the average of the $n - f$ most *clumped* gradients among the received ones (El Mhamdi et al., 2018; El-Mhamdi et al., 2020). Formally, let $\mathcal{Q} = \{g^{(1)}, g^{(2)}, ..., g^{(n)}\}$ be the set of gradients received by the parameter server and let $\mathcal{T} = \{\mathcal{V} | \mathcal{V} \subset \mathcal{Q}, |\mathcal{V}| = n - f\}$ be the set of all subsets of $\mathcal{Q}$ of cardinality $n - f$. *MDA* chooses the set

$$\mathcal{S} = \underset{\mathcal{V} \in \mathcal{T}}{\arg \min} \max_{(g^{(i)}, g^{(j)}) \in \mathcal{V}^2} \left\| g^{(i)} - g^{(j)} \right\|, \tag{9}$$

and outputs the average of the vectors in $\mathcal{S}$. El Mhamdi et al. (2018) prove that *MDA* is $(\alpha, f)$-Byzantine resilient, assuming that the following VN condition holds true:

$$\left( \frac{\sqrt{8}f}{n - f} \right)^2 \mathbb{E} \left[ \| G(\theta) - \mathbb{E}\left[ G(\theta) \right] \|^2 \right] < \| \mathbb{E}\left[ G(\theta) \right] \|^2.$$

Therefore, the multiplicative constant for *MDA* is

$$\kappa_{MDA}(n, f) = \frac{\sqrt{8}f}{n - f}. \tag{10}$$

### B.3    MEDIAN

Yin et al. (2018) introduce the *Median* aggregation rule under the assumption that $n \geq 2f + 1$. When using *Median*, the parameter server outputs the coordinate-wise median of the submitted gradients. We recall that every submitted gradient $g^{(i)} \in \mathbb{R}^d$, where $d$ is the number of parameters of the model. Formally, *Median* is defined as follows

$$Median\left( g^{(1)}, g^{(2)}, ..., g^{(n)} \right) = \begin{bmatrix} \boldsymbol{median} \left( g^{(1)}[1], g^{(2)}[1], ..., g^{(n)}[1] \right) \\ \boldsymbol{median} \left( g^{(1)}[2], g^{(2)}[2], ..., g^{(n)}[2] \right) \\ \vdots \\ \boldsymbol{median} \left( g^{(1)}[d], g^{(2)}[d], ..., g^{(n)}[d] \right) \end{bmatrix} \tag{11}$$

where $g^{(i)}[j]$ is the $j'$th coordinate of $g^{(i)}$, and ***median*** is the real-valued median. In other words,

$$\textbf{\textit{median}}\,(x_1, x_2, ..., x_n) = \arg\min_{x \in \mathbb{R}} \sum_{i=1}^{n} |x_i - x|$$

where $x_1, x_2, ..., x_n \in \mathbb{R}$. The VN condition for *Median* is the following:

$$(n - f)\,\mathbb{E}\left[\|G(\theta) - \mathbb{E}\left[G(\theta)\right]\|^2\right] < \|\mathbb{E}\left[G(\theta)\right]\|^2$$

Therefore, the multiplicative constant for *Median* is

$$\kappa_{Median}(n, f) = \sqrt{n - f}. \tag{12}$$

## B.4 BULYAN

*Bulyan* is an aggregation rule defined under the assumption that $n \geq 4f + 3$. It is actually not an aggregation rule in the conventional sense, but rather an iterative method that repetitively uses an existing GAR (El Mhamdi et al., 2018). In this paper, we use *Bulyan* on top of *Krum* defined above. Formally, *Bulyan* uses *Krum* $n - 2f - 2$ times iteratively, each time discarding the highest-scoring gradient. After that, the parameter server is left with a set $\mathcal{P}$ of the $n - 2f - 2$ "lowest-scoring" gradients selected by *Krum*, as mentioned in Appendix B.1. *Bulyan* then outputs the average of the $n - 4f - 2$ closest gradients to the coordinate-wise median of the $n - 2f - 2$ (selected) gradients $\in \mathcal{P}$.

The VN condition for *Bulyan* is the same as that of *Krum* (i.e., equation 7). Therefore, the multiplicative constant for *Bulyan* is

$$\kappa_{Bulyan}(n, f) = \sqrt{2\left(n - f + \frac{f(n - f - 2) + f^2(n - f - 1)}{n - 2f - 2}\right)}. \tag{13}$$

## C  PROOFS OMITTED FROM THE MAIN PAPER

### C.1  TECHNICAL BACKGROUND ON PRIVACY

Before demonstrating Theorem 1, we recall some classical tools from the DP literature. Below, we recall the definition of sensitivity, the privacy guarantee of the Gaussian noise injection, and the notion notion of privacy amplification by sub-sampling.

**Definition 5** (Sensitivity). *Let $f : \mathcal{X}^b \to \mathbb{R}^d$. The sensitivity of $f$, denoted by $\Delta(f)$, is the maximum norm of the difference between the outcomes of $f$ when applied on any two adjacent datasets, i.e.,*

$$\Delta(f) := \sup_{D \sim D'} \|f(D) - f(D')\|,$$

*where $D \sim D'$ denote the adjacency between the databases $D$ and $D'$ from $\mathcal{X}^b$.*

Using this notion of sensitivity, we can demonstrate that the Gaussian noise injection scheme (a.k.a. the Gaussian mechanism) satisfies $(\epsilon, \delta)$-DP for a well chosen noise injection parameter $s$.

**Lemma 1** (Dwork et al. (2014)). *Let $f : \mathcal{X}^b \to \mathbb{R}^d$, $(\epsilon, \delta) \in (0, 1)^2$, and $s > 0$. The scheme that takes $D \in \mathcal{X}^b$ as input, and outputs*

$$\mathcal{M}(D) = f(D) + y \ \text{ where } \ y \sim \mathcal{N}(0, s^2 I_d)$$

*satisfies $(\epsilon, \delta)$-DP if $s \geq \frac{\Delta(f)}{\epsilon}\sqrt{2\ln(1.25/\delta)}$.*

Finally, let us introduce the concept of privacy amplification by sub-sampling. Here, we study sub-sampling without replacement defined as follows.

**Definition 6.** *(Sub-sampling) Given a dataset $D \in \mathcal{X}^m$ and a constant $b \in [m]$, the procedure* SUB-SAMPLE$_{m \to b} : \mathcal{X}^m \to \mathcal{X}^b$ *selects $b$ points at random and without replacement from $D$.*

This sub-sampling procedure has been widely studied in the privacy preserving literature and is known to provide privacy amplification. In particular, Balle et al. (2018) demonstrated that it satisfies the following privacy amplification lemma.

**Lemma 2** (Balle et al. (2018)). *Let $\epsilon > 0$, $\delta \in (0, 1)$, $b \in [m]$, and $\mathcal{O}$ be an arbitrary output space. Let $\mathcal{M} : \mathcal{X}^b \to \mathcal{O}$ be an $(\epsilon, \delta)$-DP algorithm and $\mathcal{M}' : \mathcal{X}^m \to \mathcal{O}$ defined as $\mathcal{M}' := \mathcal{M} \circ$ SUB-SAMPLE$_{m \to b}$. Then $\mathcal{M}'$ is $(\epsilon', \delta')$-DP, with $\epsilon' = \ln\left(1 + \frac{b}{m}(e^\epsilon - 1)\right)$ and $\delta' = \frac{b\delta}{m}$.*

### C.2  PROOF OF THEOREM 1

**Theorem 1.** *Suppose that Assumption 1 holds true. Let $b \in [m]$ and $(\epsilon, \delta) \in (0, 1)^2$. Consider Algorithm 1 with $s = \frac{2C}{b\ln\left((e^\epsilon - 1)\frac{m}{b} + 1\right)}\sqrt{2\ln\left(\frac{1.25b}{m\delta}\right)}$. Then, each honest worker satisfies $(\epsilon, \delta)$-DP at every step $t \in [T]$ of the procedure.*

*Proof.* Let $t \in [T]$ be an arbitrary step of Algorithm 1 and $\theta_t$ the parameter at step $t$. Let us consider an arbitrary honest worker $w_i$. Note that the batch on which $w_i$ computes its gradient estimate is constituted of $b$ points randomly sampled without replacement from $D$. Hence we can write $B_t =$ SUB-SAMPLE$_{m \to b}(D)$. We now denote by $f : \mathcal{X}^b \to \mathbb{R}^d$ the function that evaluates the mean gradient at $\theta_t$ using $B_t$. Specifically,

$$f(B_t) := \frac{1}{b}\sum_{x \in B_t} \nabla q(\theta_t, x), \text{ for any batch } B_t \in \mathcal{X}^b. \tag{14}$$

We denote by $\mathcal{M} : \mathcal{X}^b \to \mathbb{R}^d$ the noise injection scheme, i.e., for any $B_t \in \mathcal{X}^b$,

$$\mathcal{M}(B_t) := f(B_t) + y_t; \quad y_t \sim \mathcal{N}(0, s^2 I_d). \tag{15}$$

Following the above notation, at step $t$, the honest worker $w_i$ computes the noisy gradient estimate $g_t^{(i)} = \mathcal{M} \circ$ SUB-SAMPLE$_{m \to b}(D)$. Hence, it suffices to show that $\mathcal{M} \circ$ SUB-SAMPLE$_{m \to b}$ satisfies $(\epsilon, \delta)$-DP to conclude the proof.

Since two adjacent datasets can only differ on one row, using Assumption 1, we have that $\Delta(f) \leq \frac{2C}{b}$. In particular, this implies that $s \geq \frac{\Delta(f)}{\ln\left((e^\epsilon - 1)\frac{m}{b} + 1\right)} \sqrt{2 \ln\left(\frac{1.25b}{m\delta}\right)}$. Then, according to Lemma 1, $\mathcal{M}$ is $\left(\epsilon', \frac{m}{b}\delta\right)$-DP with $\epsilon' = \ln\left((e^\epsilon - 1)\frac{m}{b} + 1\right)$. Finally, it suffices to use Lemma 2 to conclude that $\mathcal{M} \circ \text{SUB-SAMPLE}_{m \to b}$ is $(\epsilon, \delta)$-differentially private. □

## C.3 PROOF OF PROPOSITION 1

**Proposition 1.** *Let $b \in [m]$. Consider Algorithm 1 with $s > 0$. If Assumption 3 holds true, then there exists no GAR that satisfies the VN condition.*

*Proof.* Let us consider an arbitrary GAR $F$ with multiplicative constant $\kappa_F(n, f) > 0$. We denote the set of critical points of $Q$ by $\Theta^* := \{\theta \in \mathbb{R}^d \mid \nabla Q(\theta) = 0\}$. While considering Algorithm 1, the random variable that characterizes the gradients sent by the honest workers at a given parameter vector is defined as follows, for all $\theta \in \mathbb{R}^d$,

$$G(\theta) = \frac{1}{b} \sum_{x \in B} \nabla q(\theta, x) + y$$

where $B$ is a set of $b$ points randomly sampled without replacement from $D$ (denoted $B \sim D^b_{WR}$) and $y \sim \mathcal{N}(0, s^2 I_d)$. To show that the VN condition (in Definition 3) does not hold true, we show that there exists $\theta' \in \mathbb{R}^d \backslash \Theta^*$ such that

$$\kappa_F(n, f)^2 \mathbb{E}\left[\|G(\theta') - \mathbb{E}[G(\theta')]\|^2\right] \geq \|\mathbb{E}[G(\theta')]\|^2.$$

For doing so, we first observe that for any $\theta \in \mathbb{R}^d$, $G(\theta)$ is an unbiased estimator of $\nabla Q(\theta)$, i.e., $\mathbb{E}[G(\theta)] = \nabla Q(\theta)$. Furthermore, note that the injected noise $y$ is independent from the stochasticity of gradient estimate $\left(\frac{1}{b} \sum_{x \in B} \nabla q(\theta, x)\right)$. Hence, for all $\theta \in \mathbb{R}^d$,

$$\mathbb{E}\left[\|G(\theta) - \mathbb{E}[G(\theta)]\|^2\right] = \mathbb{E}_{B \sim D^b_{WR}}\left[\left\|\frac{1}{b} \sum_{x \in B} \nabla q(\theta, x) - \nabla Q(\theta)\right\|^2\right] + \mathbb{E}_{y \sim \mathcal{N}(0, s^2 I_d)}\left[\|y\|^2\right]$$

$$\geq \mathbb{E}_{y \sim \mathcal{N}(0, s^2 I_d)}\left[\|y\|^2\right] = \text{Tr}(s^2 I_d) = ds^2. \tag{16}$$

As $Q$ admits non-trivial minima, we know that $\Theta^* \neq \mathbb{R}^d$. Accordingly, there exists $\theta^* \in \Theta^*$ and $\theta' \in \mathbb{R}^d \backslash \Theta^*$. Without loss of generality, we can always take $\theta^*$ and $\theta'$ such that

$$\|\theta' - \theta^*\| \leq \frac{\kappa_F(n, f)\sqrt{d}s}{L},$$

where $L$ is the constant defined in Assumption 3. Thus, using Assumption 3 we get

$$\|\nabla Q(\theta')\| = \|\nabla Q(\theta') - \nabla Q(\theta^*)\| \leq L\|\theta' - \theta^*\| \leq \kappa_F(n, f)\sqrt{d}s. \tag{17}$$

Furthermore, thanks to (16) we know that

$$\kappa_F(n, f)^2 \mathbb{E}\left[\|G(\theta') - \mathbb{E}[G(\theta')]\|^2\right] \geq \kappa_F(n, f)^2 ds^2. \tag{18}$$

Finally, using (17) and (18) we obtain that

$$\kappa_F(n, f)^2 \mathbb{E}\left[\|G(\theta') - \mathbb{E}[G(\theta')]\|^2\right] \geq \kappa_F(n, f)^2 ds^2 \geq \|\nabla Q(\theta')\|^2 = \|\mathbb{E}[G(\theta')]\|^2.$$

The above concludes the proof. □

## C.4 PROOF OF THEOREM 2

Before we prove the theorem, we note the following implication of Assumption 2.

**Lemma 3.** *Under Assumption 2, for a given parameter $\theta$,*

$$\mathbb{E}\left[\left\|\frac{1}{b}\sum_{x\in B}\nabla q\left(\theta,\, x\right) - \nabla Q(\theta)\right\|^2\right] \leq \upsilon^2$$

*where recall that $B$ is a batch of $b$ data points chosen randomly from dataset $D$.*

*Proof.* Consider an arbitrary $B$. Then,

$$\left\|\frac{1}{b}\sum_{x\in B}\nabla q\left(\theta,\, x\right) - \nabla Q(\theta)\right\|^2 = \left\|\frac{1}{b}\sum_{x\in B}\left(\nabla q\left(\theta,\, x\right) - \nabla Q(\theta)\right)\right\|^2.$$

By triangle inequality, and the fact that $(\cdot)^2$ is a convex function, we obtain that

$$\left\|\frac{1}{b}\sum_{x\in B}\left(\nabla q\left(\theta,\, x\right) - \nabla Q(\theta)\right)\right\|^2 \leq \left(\frac{1}{b}\sum_{x\in B}\|\nabla q\left(\theta,\, x\right) - \nabla Q(\theta)\|\right)^2$$

$$\leq \frac{1}{b}\sum_{x\in B}\|\nabla q\left(\theta,\, x\right) - \nabla Q(\theta)\|^2.$$

Recall that $B$ is a set of $b$ points randomly sampled without replacement from $D$, which we denote by $B \sim D_{WR}^b$. Thus, given $\theta$,

$$\mathbb{E}\left[\left\|\frac{1}{b}\sum_{x\in B}\left(\nabla q\left(\theta,\, x\right) - \nabla Q(\theta)\right)\right\|^2\right] = \mathbb{E}_{B\sim D_{WR}^b}\left[\left\|\frac{1}{b}\sum_{x\in B}\left(\nabla q\left(\theta,\, x\right) - \nabla Q(\theta)\right)\right\|^2\right].$$

Therefore, from above we obtain that

$$\mathbb{E}\left[\left\|\frac{1}{b}\sum_{x\in B}\left(\nabla q\left(\theta,\, x\right) - \nabla Q(\theta)\right)\right\|^2\right] \leq \frac{1}{b}\mathbb{E}_{B\sim D_{WR}^b}\left[\sum_{x\in B}\|\nabla q\left(\theta,\, x\right) - \nabla Q(\theta)\|^2\right]. \tag{19}$$

Note that

$$\mathbb{E}_{B\sim D_{WR}^b}\left[\sum_{x\in B}\|\nabla q\left(\theta,\, x\right) - \nabla Q(\theta)\|^2\right] = \frac{\binom{m-1}{b-1}}{\binom{m}{b}}\sum_{x\in D}\|\nabla q\left(\theta,\, x\right) - \nabla Q(\theta)\|^2$$

$$= \frac{b}{m}\sum_{x\in D}\|\nabla q\left(\theta,\, x\right) - \nabla Q(\theta)\|^2.$$

Finally, substituting above from Assumption 2 we obtain that

$$\mathbb{E}_{B\sim D_{WR}^b}\left[\sum_{x\in B}\|\nabla q\left(\theta,\, x\right) - \nabla Q(\theta)\|^2\right] = b\upsilon^2.$$

Substitution from above in (19) concludes the proof. □

We now present the proof of Theorem 2, which is re-stated below for convenience.

**Theorem 2.** *Let $\eta \geq 0$ and $b \in [m]$. Consider Algorithm 1 with $s > 0$, a GAR $F$ satisfying the $\eta$-approximated VN condition, and $\gamma_t = 1/\sqrt{t}$ for all $t \in [T]$. If Assumptions 1, 2, and 3 hold true, then there exists $\alpha \in [0,\, \pi/2)$ and $\mu \in [0,\, \infty)$ such that for any $T \geq 1$,*

$$\min_{t\in[T]}\mathbb{E}\left[\|\nabla Q(\theta_t)\|^2\right] \leq \max\left\{\eta^2,\, \frac{Q(\theta_1) - Q^*}{(1-\sin\alpha)}\left(\frac{1}{\sqrt{T}}\right) + \frac{\mu\sigma^2 L}{2(1-\sin\alpha)}\left(\frac{1+\ln T}{\sqrt{T}}\right)\right\}$$

*where $Q^*$ is the minimum value of $Q$, i.e., $Q^* = \min\limits_{\theta\in\mathbb{R}^d} Q(\theta)$, and $\sigma = \sqrt{\upsilon^2 + ds^2 + C^2}$.*

*Proof.* Recall from (4) in Algorithm 1 that, for all $t$,

$$\theta_{t+1} = \theta_t - \gamma_t R_t$$

where $R_t = F\left(\tilde{g}_t^{(1)}, \ldots, \tilde{g}_t^{(n)}\right)$. Thus, under Assumption 3, we obtain that for all $t$,

$$Q(\theta_{t+1}) \leq Q(\theta_t) - \gamma_t \langle \nabla Q(\theta_t), R_t \rangle + \frac{\gamma_t^2 L}{2} \|R_t\|^2. \tag{20}$$

We denote by $\mathbb{E}_t[\cdot]$ the conditional expectation given the history $\mathcal{P}_t = \{\theta_1, \ldots, \theta_t; R_1, \ldots, R_{t-1}\}$. By taking the conditional expectation $\mathbb{E}_t[\cdot]$ on both sides in (20) we obtain that

$$\mathbb{E}_t\left[Q(\theta_{t+1})\right] \leq Q(\theta_t) - \gamma_t \langle \nabla Q(\theta_t), \mathbb{E}_t[R_t] \rangle + \frac{\gamma_t^2 L}{2} \mathbb{E}_t\left[\|R_t\|^2\right]. \tag{21}$$

Recall from (5) that, for all $t$,

$$G(\theta_t) := \frac{1}{b} \sum_{x \in B_t} \nabla q(\theta_t, x) + y_t$$

where $B_t$ is a set of $b$ points randomly sampled without replacement from $D$ and $y_t \sim \mathcal{N}(0, s^2 I_d)$. By Definition 4 of the $\eta$-approximated VN condition, when $\|\mathbb{E}_t[G(\theta_t)]\| > \eta$,

$$\kappa_F(n, f)^2 \mathbb{E}_t\left[\|G(\theta_t) - \mathbb{E}_t[G(\theta_t)]\|^2\right] < \|\mathbb{E}_t[G(\theta_t)]\|^2. \tag{22}$$

Now, consider an arbitrary integer $T \geq 1$. We can partition set $[T] = \{1, \ldots, T\}$ into two sets $S_T$ and $\overline{S}_T$ such that

$$S_T = \{t \in T \mid \|\mathbb{E}_t[G(\theta_t)]\| \leq \eta\}, \quad \text{and} \quad \overline{S}_T = [T] \setminus S_T. \tag{23}$$

First, we consider the case when $S_T \neq \emptyset$. In this case, $\exists t^* \in [T]$ such that $\|\mathbb{E}_{t^*}[G(\theta_{t^*})]\| \leq \eta$. As $\mathbb{E}[G(\theta)] = \nabla Q(\theta)$ for any given $\theta$, this implies that $\|\nabla Q(\theta_{t^*})\| \leq \eta$. Thus,

$$\min_{t \in [T]} \|\nabla Q(\theta_t)\| \leq \eta.$$

Thus, the theorem is trivially true in this case. Next, we consider the case when $S_T = \emptyset$, i.e., $\overline{S}_T = [T]$. Thus, in this case,

$$\|\mathbb{E}_t[G(\theta_t)]\| > \eta, \quad \forall t \in [T].$$

This implies that (22) holds true for all $t \in [T]$. Thus, from prior results in Blanchard et al. (2017); El Mhamdi et al. (2018), we obtain that GAR $F$ is $(\alpha, f)$-Byzantine resilient with respect to $G(\theta_t)$ for each $t \in [T]$. Therefore, by Definition 2, there exists $\alpha \in [0, \pi/2)$ and $\mu < \infty$ such that (recall that $\mathbb{E}[G(\theta)] = \nabla Q(\theta)$ for any given $\theta$)

$$\langle \nabla Q(\theta_t), \mathbb{E}_t[R_t] \rangle \geq (1 - \sin\alpha) \|\nabla Q(\theta_t)\|^2, \quad \text{and} \quad \mathbb{E}_t\left[\|R_t\|^2\right] \leq \mu \mathbb{E}_t\left[\|G(\theta_t)\|^2\right], \quad \forall t \in [T].$$

Substituting from above in (21) we obtain that, for all $t \in [T]$,

$$\mathbb{E}_t\left[Q(\theta_{t+1})\right] \leq Q(\theta_t) - \gamma_t(1 - \sin\alpha) \|\nabla Q(\theta_t)\|^2 + \frac{\gamma_t^2 L\mu}{2} \mathbb{E}_t\left[\|G(\theta_t)\|^2\right]. \tag{24}$$

Under Assumptions 1 and 2, from Lemma 3 we obtain that, for all $t \in [T]$,

$$\mathbb{E}_t\left[\|G(\theta_t)\|^2\right] \leq \mathbb{E}_t\left[\|G(\theta_t) - \mathbb{E}_t[G(\theta_t)]\|^2\right] + \|\nabla Q(\theta_t)\|^2 \leq (v^2 + ds^2) + C^2.$$

Recall that $\sigma^2 := v^2 + ds^2 + C^2$. Substituting from above in (24) we obtain that

$$\mathbb{E}_t\left[Q(\theta_{t+1})\right] \leq Q(\theta_t) - \gamma_t(1 - \sin\alpha) \|\nabla Q(\theta_t)\|^2 + \frac{\sigma^2 L\mu}{2} \gamma_t^2, \quad \forall t \in [T].$$

As $\mathbb{E}[\cdot] = \mathbb{E}_1[\ldots \mathbb{E}_t[\cdot] \ldots]$, from above we obtain that

$$\mathbb{E}\left[Q(\theta_{t+1})\right] \leq \mathbb{E}[Q(\theta_t)] - \gamma_t(1 - \sin\alpha) \mathbb{E}\left[\|\nabla Q(\theta_t)\|^2\right] + \frac{\sigma^2 L\mu}{2} \gamma_t^2, \quad \forall t \in [T].$$

Thus, by taking summation on both sides over all $t \in [T]$, we obtain that

$$\mathbb{E}\left[Q(\theta_{t+1})\right] \leq \mathbb{E}\left[Q(\theta_1)\right] - (1 - \sin\alpha) \sum_{t=1}^{T} \gamma_t \mathbb{E}\left[\|\nabla Q(\theta_t)\|^2\right] + \frac{\sigma^2 L\mu}{2} \sum_{t=1}^{T} \gamma_t^2.$$

As $\theta_1$ is a priori fixed, $\mathbb{E}\left[Q(\theta_1)\right] = Q(\theta_1)$. Upon subtracting $Q^*$ on both sides, and noting that $\mathbb{E}\left[Q(\theta)\right] \geq Q^*$ for all $\theta$, we obtain that

$$0 \leq \mathbb{E}\left[Q(\theta_{t+1})\right] - Q^* \leq Q(\theta_1) - Q^* - (1 - \sin\alpha) \sum_{t=1}^{T} \gamma_t \mathbb{E}\left[\|\nabla Q(\theta_t)\|^2\right] + \frac{\sigma^2 L\mu}{2} \sum_{t=1}^{T} \gamma_t^2.$$

By re-arranging,

$$(1 - \sin\alpha) \sum_{t=1}^{T} \gamma_t \mathbb{E}\left[\|\nabla Q(\theta_t)\|^2\right] \leq \mathbb{E}\left[Q(\theta_1)\right] - Q^* + \frac{\sigma^2 L\mu}{2} \sum_{t=1}^{T} \gamma_t^2.$$

Upon substituting $\gamma_t := 1/\sqrt{t}$, we obtain that

$$(1 - \sin\alpha) \frac{\sum_{t=1}^{T} \mathbb{E}\left[\|\nabla Q(\theta_t)\|^2\right]}{\sqrt{T}} \leq \mathbb{E}\left[Q(\theta_1)\right] - Q^* + \frac{\sigma^2 L\mu}{2} \sum_{t=1}^{T} \frac{1}{t}.$$

Note that $\sum_{t=1}^{T} \frac{1}{t} \leq \ln T + 1$. Thus, by multiplying both sides by $1/\sqrt{T}$, we obtain that

$$(1 - \sin\alpha) \frac{\sum_{t=1}^{T} \mathbb{E}\left[\|\nabla Q(\theta_t)\|^2\right]}{T} \leq \frac{Q(\theta_1) - Q^*}{\sqrt{T}} + \frac{\sigma^2 L\mu}{2} \left(\frac{\ln T + 1}{\sqrt{T}}\right).$$

Hence, as $\min_{t \in [T]} \mathbb{E}\left[\|\nabla Q(\theta_t)\|^2\right] \leq \left(\frac{1}{T}\right) \sum_{t=1}^{T} \mathbb{E}\left[\|\nabla Q(\theta_t)\|^2\right]$,

$$(1 - \sin\alpha) \min_{t \in [T]} \mathbb{E}\left[\|\nabla Q(\theta_t)\|^2\right] \leq \frac{Q(\theta_1) - Q^*}{\sqrt{T}} + \frac{\sigma^2 L\mu}{2} \left(\frac{\ln T + 1}{\sqrt{T}}\right).$$

Hence, the proof. $\qquad \square$

## C.5 Proof of Proposition 2

**Proposition 2.** *Let $b \in [m]$, $(\epsilon, \delta) \in (0, 1)^2$. Consider Algorithm 1 with privacy budget $(\epsilon, \delta)$ and GAR $F$ with multiplicative constant $\kappa_F(n, f) > 0$. Then the following assertions hold true.*

1. *Under Assumptions 1 and 3, the $\eta$-approximated VN condition holds true only if*

$$\eta^2 \geq 4\kappa_F(n, f)^2 C^2 d \ln\left(\frac{1.25b}{m\delta}\right) \frac{1}{bm(e^\epsilon - 1)}.$$

2. *Additionally under Assumption 2, if*

$$\eta^2 \geq \kappa_F(n, f)^2 \left(8C^2 d \ln\left(\frac{1.25b}{m\delta}\right) \left(\frac{1}{m(e^\epsilon - 1)} + \frac{1}{b}\right)^2 + \upsilon^2\right)$$

*then $F$ satisfies the $\eta$-approximated VN condition.*

*Proof.* Let us consider an arbitrary GAR $F$ with multiplicative constant $\kappa_F(n, f) > 0$. We denote the set of points that do not satisfy the constraint on the gradient of $Q$ by $\Theta_\eta := \{\theta \in \mathbb{R}^d \mid \|\nabla Q(\theta)\| \leq \eta\}$. While considering Algorithm 1, the random variable that characterizes the gradients sent by the honest workers is defined as follows:

$$G(\theta) := \frac{1}{b} \sum_{x \in B} \nabla q(\theta, x) + y, \ \forall \theta \in \mathbb{R}^d$$

where $B$ is a set of $b$ points randomly sampled without replacement from $D$ (denoted $B \sim D_{WR}^b$) and $y \sim \mathcal{N}(0, s^2 I_d)$. As in the proof of Proposition 1, we note that for all $\theta \in \mathbb{R}^d$, $G(\theta)$ is an unbiased estimate of $\nabla Q(\theta)$, hence we also have that $\Theta_\eta := \{\theta \in \mathbb{R}^d \mid \|\mathbb{E}[G(\theta)]\| \leq \eta\}$. Furthermore, the injected noise is independent of the stochasticity of gradients. Hence for all $\theta \in \mathbb{R}^d$,

$$\mathbb{E}\left[\|G(\theta) - \mathbb{E}[G(\theta)]\|^2\right] = \mathbb{E}_{B \sim D_{WR}^b}\left[\left\|\frac{1}{b}\sum_{x \in B} \nabla q(\theta, x) - \nabla Q(\theta)\right\|^2\right] + ds^2 \qquad (25)$$

**1.** We now demonstrate a necessary condition for $F$ to satisfy the $\eta$-approximated VN condition under Assumptions 1 and 3. To do so, we first show reductio ad absurdum that the $\eta$-approximated VN condition holds true only if

$$\eta^2 \geq \kappa_F(n, f)^2 d\left(\frac{8C^2 \ln\left(\frac{1.25b}{m\delta}\right)}{b^2 \ln\left((e^\epsilon - 1)\frac{m}{b} + 1\right)^2}\right).$$

Let us assume that there exists $\eta > 0$ such that

$$\eta^2 < \kappa_F(n, f)^2 d\left(\frac{8C^2 \ln\left(\frac{1.25b}{m\delta}\right)}{b^2 \ln\left((e^\epsilon - 1)\frac{m}{b} + 1\right)^2}\right), \qquad (26)$$

and such that the $\eta$-approximated VN condition holds. That is, for all $\theta \in \mathbb{R}^d \backslash \Theta_\eta$,

$$\kappa_F(n, f)^2 \mathbb{E}\left[\|G(\theta) - \mathbb{E}[G(\theta)]\|^2\right] < \|\mathbb{E}[G(\theta)]\|^2. \qquad (27)$$

Let us first note that from (25), replacing $s^2$ according to the privacy budget from Theorem 1, we have for any $\theta \in \mathbb{R}^d$

$$d\left(\frac{8C^2 \ln\left(\frac{1.25b}{m\delta}\right)}{b^2 \ln\left((e^\epsilon - 1)\frac{m}{b} + 1\right)^2}\right) \leq \mathbb{E}\left[\|G(\theta) - \mathbb{E}[G(\theta)]\|^2\right]. \qquad (28)$$

Hence, combining (27) and (28), we get that for all $\theta \in \mathbb{R}^d \backslash \Theta_\eta$,

$$\kappa_F(n, f)^2 d\left(\frac{8C^2 \ln\left(\frac{1.25b}{m\delta}\right)}{b^2 \ln\left((e^\epsilon - 1)\frac{m}{b} + 1\right)^2}\right) < \|\mathbb{E}[G(\theta)]\|^2. \qquad (29)$$

Recall that $Q$ admits a local minima, hence there exists $\theta^*$ such that $\|\nabla Q(\theta^*)\| = 0$. Furthermore, under Assumption 3, we know that $\|\nabla Q(.)\|^2$ is a continuous on $\mathbb{R}^d$; hence its range on $\mathbb{R}^d$ includes the interval $\left[0, \ \kappa_F(n, f)^2 d\left(\frac{8C^2 \ln\left(\frac{1.25b}{m\delta}\right)}{b^2 \ln\left((e^\epsilon - 1)\frac{m}{b} + 1\right)^2}\right)\right]$. In particular, there exist $\theta' \in \mathbb{R}^d$ such that

$$\|\nabla Q(\theta')\|^2 = \|\mathbb{E}[G(\theta')]\|^2 = \kappa_F(n, f)^2 d\left(\frac{8C^2 \ln\left(\frac{1.25b}{m\delta}\right)}{b^2 \ln\left((e^\epsilon - 1)\frac{m}{b} + 1\right)^2}\right). \qquad (30)$$

On the one hand, assuming that $\theta' \in \Theta_\eta$, leads to a contradiction with (26). On the other hand, assuming that $\theta' \in \mathbb{R}^d \backslash \Theta_\eta$, leads to another contradiction with (29); hence also with (26). Finally, we obtain reductio ad absurdum that the $\eta$-approximated VN condition holds only if

$$\eta^2 \geq \kappa_F(n, f)^2 d\left(\frac{8C^2 \ln\left(\frac{1.25b}{m\delta}\right)}{b^2 \ln\left((e^\epsilon - 1)\frac{m}{b} + 1\right)^2}\right). \qquad (31)$$

If we further note that for any $x \in \mathbb{R}_{\geq 0}$, $\ln(x + 1) \leq 2\sqrt{x}$ we have

$$\kappa_F(n, f)^2 d\left(\frac{8C^2 \ln\left(\frac{1.25b}{m\delta}\right)}{b^2 \ln\left((e^\epsilon - 1)\frac{m}{b} + 1\right)^2}\right) \geq 4\kappa_F(n, f)^2 d\left(\frac{C^2 \ln\left(\frac{1.25b}{m\delta}\right)}{b^2(e^\epsilon - 1)\frac{m}{b}}\right). \qquad (32)$$

Finally, combining (31) and (32) proves the necessary condition.

**2.** We now show the sufficient condition for $F$ to satisfy the $\eta$-approximated VN condition by further supposing that Assumption 2 holds true. Let us consider

$$\eta^2 \geq \kappa_F(n,f)^2 \left( 8C^2 d \ln\left(\frac{1.25b}{m\delta}\right) \left(\frac{1}{m(e^\epsilon-1)} + \frac{1}{b}\right)^2 + \upsilon^2 \right) \qquad (33)$$

Note that for any $x \in \mathbb{R}_{>0}$, $1 - \frac{1}{x} \leq \ln(x)$. Accordingly, we have

$$\frac{1}{m(e^\epsilon-1)} + \frac{1}{b} = \frac{1}{b}\left(1 - \frac{1}{\frac{m}{b}(e^\epsilon-1)+1}\right)^{-1} \geq \frac{1}{b}\left(\ln\left(\frac{m}{b}(e^\epsilon-1)+1\right)\right)^{-1} \qquad (34)$$

By combining (33) and (34), we get

$$\eta^2 \geq \kappa_F(n,f)^2 \left( d\left(\frac{8C^2 \ln\left(\frac{1.25b}{m\delta}\right)}{b^2 \left(\ln\left((e^\epsilon-1)\frac{m}{b}+1\right)\right)^2}\right) + \upsilon^2 \right). \qquad (35)$$

Furthermore, by Lemma 3, under Assumption 2, we have that

$$\mathbb{E}\left[\|G(\theta) - \mathbb{E}\left[G(\theta)\right]\|^2\right] \leq \upsilon^2 + d\left(\frac{8C^2 \ln\left(\frac{1.25b}{m\delta}\right)}{b^2 \left(\ln\left((e^\epsilon-1)\frac{m}{b}+1\right)\right)^2}\right).$$

Thus,

$$\kappa_F(n,f)^2 \,\mathbb{E}\left[\|G(\theta) - \mathbb{E}\left[G(\theta)\right]\|^2\right] \leq \kappa_F(n,f)^2 \left(\upsilon^2 + d\left(\frac{8C^2 \ln\left(\frac{1.25b}{m\delta}\right)}{b^2 \left(\ln\left((e^\epsilon-1)\frac{m}{b}+1\right)\right)^2}\right)\right).$$

Finally, by using (35), for any $\theta \in \mathbb{R}\backslash\Theta_\eta$ we have

$$\kappa_F(n,f)^2 \,\mathbb{E}\left[\|G(\theta) - \mathbb{E}\left[G(\theta)\right]\|^2\right] \leq \eta^2 < \|\mathbb{E}\left[G(\theta)\right]\|^2.$$

The above concludes the proof.

$\square$

## C.6 Proof of Corollary 1

**Corollary 1.** *Let $b \in [m]$, $(\epsilon,\delta) \in (0,1)^2$. Consider Algorithm 1 with privacy budget $(\epsilon,\delta)$ and GAR F, and $\gamma_t = 1/\sqrt{t}$ for all $t \in [T]$. If Assumptions 1, 2, and 3 are satisfied, then for any $T \geq 1$,*

$$\min_{t \in [T]} \mathbb{E}\left[\|\nabla Q(\theta_t)\|^2\right] \leq \max\left\{ \kappa_F(n,f)^2 \left(8C^2 d \ln\left(\frac{1.25b}{m\delta}\right)\left(\frac{1}{m(e^\epsilon-1)}+\frac{1}{b}\right)^2 + \upsilon^2\right), O\left(\frac{\ln T}{\sqrt{T}}\right)\right\}.$$

*Proof.* Under the stated assumptions, we note from Proposition 2 that GAR $F$ satisfies the $\eta$-approximated VN condition when

$$\eta \geq \kappa_F(n,f)^2 \left(8C^2 d \ln\left(\frac{1.25b}{m\delta}\right)\left(\frac{1}{m(e^\epsilon-1)}+\frac{1}{b}\right)^2 + \upsilon^2\right).$$

Thus, Theorem 2 implies that there exist $\alpha \in [0, \pi/2)$ and $\mu \in [0, \infty)$ such that for any $T \geq 1$,

$$\min_{t \in [T]} \mathbb{E}\left[\|\nabla Q(\theta_t)\|^2\right] \leq$$

$$\max\left\{ \kappa_F(n,f)^2 \left(8C^2 d \ln\left(\frac{1.25b}{m\delta}\right)\left(\frac{1}{m(e^\epsilon-1)}+\frac{1}{b}\right)^2 + \upsilon^2\right), \frac{Q(\theta_1)-Q^*}{(1-\sin\alpha)}\left(\frac{1}{\sqrt{T}}\right) + \frac{\mu\sigma^2 L}{2(1-\sin\alpha)}\left(\frac{1+\ln T}{\sqrt{T}}\right)\right\}.$$

Finally, note that

$$\frac{Q(\theta_1)-Q^*}{(1-\sin\alpha)}\left(\frac{1}{\sqrt{T}}\right) + \frac{\mu\sigma^2 L}{2(1-\sin\alpha)}\left(\frac{1+\ln T}{\sqrt{T}}\right) \in O\left(\frac{\ln T}{\sqrt{T}}\right).$$

Hence, the proof. $\square$

# D ADDITIONAL INFORMATION ON THE EXPERIMENTAL SETUP

## D.1 MOMENTUM SGD

While the wish for simplicity of presentation has led us to focus on SGD in the paper, our use of momentum in the experiments is motivated by the following two main observations.

1. El-Mhamdi et al. (2021) recently demonstrated empirically that momentum at the workers can improve the resilience of the system to Byzantine workers.

2. Momentum has been demonstrated to improve the convergence rate of SGD by reducing the variance of stochastic gradients. In the context of Algorithm 1, momentum can be shown to reduce the variance of the noise injected by the workers, while preserving the privacy budget.

## D.2 PRIVACY BUDGET AND NUMBER OF TRAINING STEPS

The values of per step and per worker privacy budget $\epsilon \in \{0.2, 0.1, 0.05\}$ and $\delta = 10^{-5}$ that we consider in Section 4.1 yield, respectively, an overall privacy budget of $\overline{\epsilon} \in \{7.58, 4.77, 3.46\}$ and $\overline{\delta} = 1.2 \times 10^{-4}$ for every batch size considered, after running the algorithm for $T$ steps. Since $(\overline{\epsilon}, \overline{\delta})$ directly depends on the batch size as well as the total number of learning steps $T$, we adapt the value of $T$ for every run in order to obtain the same $(\overline{\epsilon}, \overline{\delta})$ in all experiments (regardless of the batch size used). The maximum value of $T$ considered is 300 steps for the smallest batch size $b = 25$. Larger batch sizes have $T < 300$ [2]. Finally, given a fixed batch size and number of steps, we evaluate the overall privacy budgets using the PyTorch package Opacus that is based on the *moments accountant* algorithm (Abadi et al., 2016), as well as the advanced composition theorem (Kairouz et al., 2015).

## D.3 STUDIED ATTACKS

In the experiments of this paper, we use two state-of-the-art attacks that we refer to as *little* and *empire*. Both attacks rely on the same core idea. Let $\zeta$ be fixed a non-negative real number and let $a_t$ be the attack vector at time step $t$. At every time step $t$, all Byzantine workers send $\overline{g_t} + \zeta a_t$ to the server, where $\overline{g_t}$ is an approximation of the real gradient $G(\theta_t)$ at step $t$. The specific details of each attack are mentioned below.

**Little is Enough (Baruch et al., 2019).** In this attack, $a_t = -\sigma_t$, where $\sigma_t$ is the opposite vector of the coordinate-wise standard deviation of $G(\theta_t)$. In our experiments, we set $\zeta = 1$ for *little*.

**Fall of Empires (Xie et al., 2019).** In this attack, $a_t = -\overline{g_t}$. All Byzantine workers thus send $(1 - \zeta)\overline{g_t}$ at step $t$. In our experiments, we set $\zeta = 1.1$ for *empire*, corresponding to $\epsilon = 0.1$ in the notation of the original paper.

# E ADDITIONAL EXPERIMENTAL RESULTS

We remind the reader that all experiments are reproducible in one command (refer to `code/README.md`). All plots can be found in `plots/` in the supplementary folder. Due to space limitations, `plots/` only includes the plots for the Fashion-MNIST dataset. The plots corresponding to MNIST can also be generated by running the code.

## E.1 SANITY CHECKS

**Impact of Batch Size in Non-Private Setting.** Hereafter, we provide some additional results on the impact of the batch size in the settings where DP is not enforced. These results validate our claim on the mild impact of the batch size and the GAR selection when no noise is injected at the worker level.

In Figure 3, we study the impact of the batch size on the learning accuracy in a non-private setting where the workers do not care about leaking private information and thus share their gradients in the clear without adding noise to them. Unlike in Figure 2 from Section 4.2, all curves in all four

---

[2]Note that to avoid technical difficulties, we actually run 300 learning steps for all experiments. However, we report the maximum cross-accuracies at the end of $T \leq 300$ steps, where $T$ is defined as above.

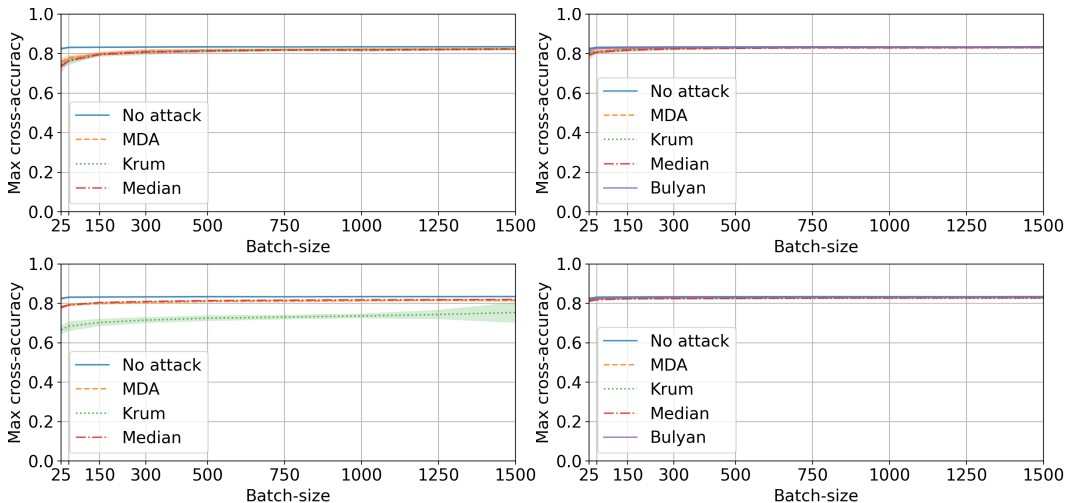

Figure 3: Impact of the batch size on the cross-accuracy of the learning on Fashion-MNIST in the non-private scenario. The first and second rows show the *little* and *empire* attacks respectively. The first and second columns correspond to $f = 6$ and $f = 3$ Byzantine nodes respectively.

threat scenarios are mostly flat. This indicates that the batch size and the GAR indeed become crucial factors in the HPO process whenever DP and BR are combined, and has a negligible impact when only one of these notions is enforced (refer to Figure 3 and the *No attack* curves in Figure 2).

**Impact of $\epsilon$ with and without Byzantine workers.** We also provide below some results studying the impact of varying the privacy parameter $\epsilon$ alone under different attack scenarios. As shown in Figure 4, in the non-Byzantine setting (No attack), increasing $\epsilon$ (i.e., weakening the privacy guarantees) does increase the maximum cross-accuracy encountered in the learning, but at a negligible rate. All *No attack* curves only show mild improvement in the cross-accuracy compared to the GARs under attacks. This means that the standard privacy-accuracy trade-off in DP is quite mild in our experiments. However, when there are Byzantine workers in the system executing state-of-the-art attacks, the impact of $\epsilon$ on the learning accuracy is more important, especially in the case of *little* (first row of Figure 4). In some instances where the executed attack might not be strong (e.g., *empire* and $f = 3$), the impact of $\epsilon$ is mild even in the Byzantine setting.

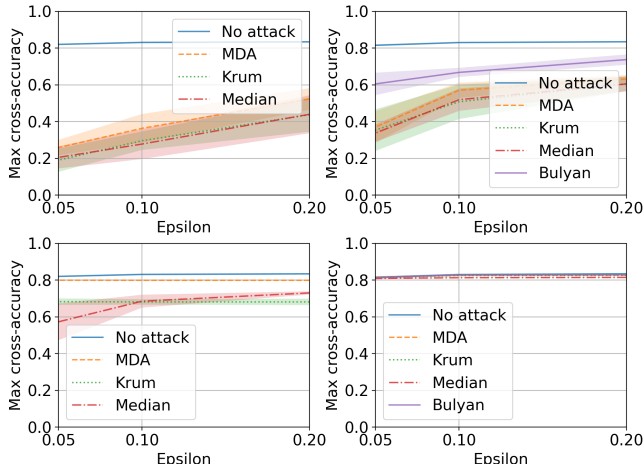

Figure 4: Impact of $\epsilon$ on the cross-accuracy of the learning on Fashion-MNIST for $b = 1000$. The first and second rows show the *little* and *empire* attacks respectively. The first and second columns correspond to $f = 6$ and $f = 3$ Byzantine nodes respectively.

### E.2 Satisfiability of the VN Condition in Practice

Verifying whether the VN condition (or the $\eta$-approximated VN condition) holds is actually impossible in practice as we do not have access to the real gradient distribution of the honest workers $G(.)$. Instead we estimate statistics on $G(.)$ by using the gradient estimates of the honest workers at every step of the learning. Furthermore, to facilitate the visualization, we actually compute the VN ratio defined below.

**VN Ratio:** We re-arrange the terms of the $\eta$-approximated VN condition as follows:

$$\kappa_F(n, f)^2 \, \mathbb{E}\left[\|G(\theta) - \mathbb{E}\left[G(\theta)\right]\|^2\right] < \|\mathbb{E}\left[G(\theta)\right]\|^2 \tag{36}$$

$$\Leftrightarrow \underbrace{\frac{\sqrt{\mathbb{E}\left[\|G(\theta) - \mathbb{E}\left[G(\theta)\right]\|^2\right]}}{\|\mathbb{E}\left[G(\theta)\right]\|}}_{\text{VN ratio}} < \frac{1}{\kappa_F(n, f)}. \tag{37}$$

In the following, we refer to the left hand side of (37) as the *VN ratio*. This is the ratio of the standard deviation of the honest gradients to the norm of the expected honest gradient, which has already been discussed by El Mhamdi et al. (2018). In simulations, the VN ratio can be measured during the training along the parameter trajectory by estimating the variance and the expectation of the gradients sent by the honest workers. For a GAR $F$ for which the constant $\kappa_F(n, f)$ is known (e.g., see Appendix B), we can measure the VN ratio to get insights on the satisfiability of the $\eta$-approximated VN condition.

**VN Ratio in Our Experiments: Takeaways.** In Figure 5, we show the measured VN ratios along the parameter trajectory $(\theta_1, \ldots, \theta_{300})$, for both the gradients *sampled* by each honest worker and the gradients *submitted* to the GAR. The *sampled* honest gradients are the gradients computed by the honest workers using the data batches sampled in each step of the training. The *submitted* honest gradients are the gradients submitted by the honest workers to the parameter server, after having corrupted them using additive Gaussian noise and having applied momentum.

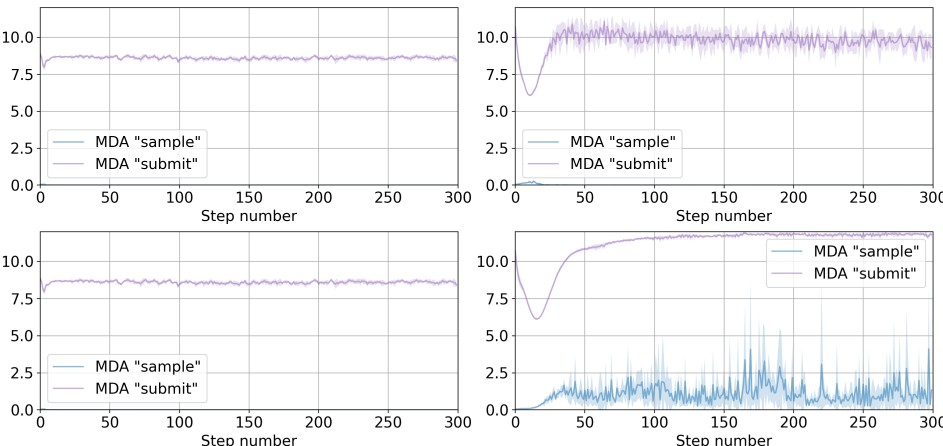

Figure 5: VN ratios for *MDA* and $b = 1000$, under the same settings (and layout) as in Figure 2. "Sample" corresponds to the VN ratio of the sampled gradients, i.e., $g_t^{(i)}$ for the honest worker $i$. "Submit" corresponds to the VN ratio observed by the GAR. The latter significantly differs from the former mostly due to the added Gaussian noise for DP.

Figure 5 summarizes the value of the VN ratio we encountered during the learning under the same settings (and layout) as in Figure 2. We observe that the VN ratio of a *sampled* gradient can be extremely small compared to the VN ratio of its corresponding *submitted* gradient. Our first takeaway regarding the *submitted* gradient is that its VN ratio is "nearly" constant. As the VN ratio of the privacy

noise is constant[3], the aforementioned observation suggests that the privacy noise is substantially larger than the sampled gradients; hence dominating the VN ratio of the *submitted* gradients.

Recalling that $\kappa_{MDA}(n, f) = \sqrt{8}f/(n-f)$, Figure 5 shows that if (in our experiments) $\kappa_{MDA}$ where to be smaller than $0.08$, the $\eta$-approximated VN condition would have been satisfied during the entire training. This is not the case in our setting where $n = 15$. This brings us to our second takeaway, discussed as follows. For *MDA*, there exists extreme (but arguably reasonable) settings for which the VN condition can actually be satisfied, for at least several hundred training steps. However, we did not have to use such extreme settings in our experiments to observe convergence.

---

[3]The VN ratio can be treated as a property of a generic random vector, e.g., stochastic gradient or privacy noise. As the distribution of the privacy noise does not change during training, its VN ratio is constant.

