# OpenReview forum: "Combining Differential Privacy and Byzantine Resilience in Distributed SGD"
_ICLR.cc/2022/Conference — ICLR 2022 Submitted_

### Official Review · Reviewer_NXde · 2021-11-01

**Correctness:** 3
**Technical Novelty And Significance:** 2
**Empirical Novelty And Significance:** 2
**Recommendation:** 3
**Confidence:** 3

**Main Review:**

Strengths:

-Proposition 1 and (especially) Proposition 2 are interesting

-The problem is interesting

Weaknesses:

General points:

-Measuring privacy loss in terms of privacy loss per iteration is not meaningful. Privacy loss accumulates, so it is very unusual and not informative to measure the loss of each step; instead epsilon should capture the privacy loss of the full algorithm (T rounds) to quantify the risk of inference attacks on the model

-The significance of the VN condition is not adequately explained. How exactly is this condition relevant to BR?

-The convergence results are not very interpretable, seem to have incorrect units

-Writing needs a lot of work

Specifics:
0. Abstract

-"...rendered invalid when workers enforce DP" is not accurate for two reasons: you only show this for a particular DP protocol, namely the Gaussian mechanism; showing it fundamentally for any DP mechanism would be much more interesting. Also, what you show is in terms of the VN condition, which is not explicitly connected to BR.

1. Introduction

-"trusted server" and "honest but curious server" are both used to describe the set up, which is confusing. Usually "trusted server" means honest and not curious (i.e. not a privacy threat).

-"Byzantine" should be informally defined or briefly described early on before it is used many times

-Dwork et al (2014) is a strange choice for citation for a sentence about private ML. There are many other more relevant works from 2014 to present. Also, "especially when considering neural networks" does not seem to add anything and I'm not sure I agree that neural nets are special in this sense.

-Abadi et al (2016) is the wrong/incomplete citation for DP SGD: Bassily et al (2014) and Song et al (2013) considered DP SGD earlier.

-Theorem (Informal): should state conditions on loss; need to define (or informally explain, at the very least) "approximated VN" before using it in a theorem; paragraph following Theorem is too much detail for not having defined VN

-"parameters have very little impact in most settings when considering DP or BR separately" is not clear

-Figure 1. Is the number of iterations fixed?

2. Problem Settign

-"common dataset" what does this mean? Are the m points divided/partitioned among n workers or all workers have access to D?

-"*By far, the most widely used* approach....is *the* differentially private version...": strong claim made without any evidence; also, there are many ways to provide DP besides Gaussian noise, so there is no single "the" DP version of SGD.

-**"we are mainly interested in...per-step...privacy": Why?! This is not nearly as meaningful. If T goes to infinity then essentially the algorithm provides no privacy at all but your epsilon might still be small, which is very misleading.**

-Def 2: explain intuition; provide an example of a BR GAR

Section 3:

-Algorithm 1: clarify the presentation. When you loop through "honest" and "Byzantine" workers, it seems as if the analyst/curator who is implementing the algorithm knows which workers are honest and not, which is clearly not the case

-Contextualize Theorem 1. the privacy properties of Gaussian mechanism and subsampling are well-known, so this theorem is not at all novel; this should be stated. Also the log term in the denominator can be tightened; can take $s^2 \approx C^2 log(1/\delta)/\epsilon^2 m^2$

-**What's the significance of VN condition?** How does it relate to BR?

-"when $\epsilon$ and $\delta$ are non-zero...": strange sentence because $s$ increases as $\epsilon$ and $\delta$ decrease.

-Theorem 2: why doesn't $\kappa$ appear? Units appear to be wrong. Should provide comparison to Aliastarh, Allen-Zhu, Li (2018) and the references therein. Also should compare to DP optimization rates

-Corollary 1: misleading because $\epsilon$ is not the actual privacy budget of full T-round algorithm, so first term should also scale with T. This remark applies to experiments too.

**Summary Of The Paper:**

The paper considers the problem of distributed learning via SGD when a fraction of workers are Byzantine and the rest want their data to be kept private. The authors consider a generic/naiive combination of Byzantine resilience (BR) and differentially privacy (DP) in Algorithm 1. They show that the VN condition is incompatible with Gaussian noised SGD, but propose an approximate VN conidtion that can be realized by noisy SGD. They then establish a convergence guarantee for the algorithm and conduct numerical experiments.

**Summary Of The Review:**

The paper makes some good progress towards an understanding of DP and BR in SGD via Proposition 1 and 2. However, the convergence results are not very clean and not properly contextualized. The writing is rather poor and the $\epsilon$ issue is a big one. I cannot recommend acceptance in the current form.

---

> ### Author Response · Authors · 2021-11-12
> **Response to Reviewer NXde (1/2)**
>
> We thank the reviewer for their comments and provide detailed answers to their concerns below.
>
> **Comments on the abstract:**
> - We only focus on the workers enforcing DP via the Gaussian mechanism, which is arguably the most standard DP mechanism in machine learning (off-the-shelf libraries implement this scheme by default, e.g.,  [TensorFlow Privacy](https://github.com/tensorflow/privacy/tree/master/tensorflow_privacy), [Opacus (PyTorch)](https://github.com/pytorch/opacus). This specification is explicitly stated in our introduction and model setting. Hence, a general analysis of the integration of DP and BR without specifying the DP mechanism, even though interesting, is out of the scope of this paper.
>
> - Regarding the link between VN condition and BR, please refer to our answer in below on the significance of the VN condition.
>
> **Comments on the introduction:**
>
> - As explained in our threat model in Section 2, our analysis considers the server to be "honest-but-curious". We only use the notion of "trusted server" in the first paragraph of the introduction to explain how distributed SGD works in the vanilla setting (i.e., without privacy and without Byzantine workers).
>
> - We will provide an intuitive explanation of what Byzantine means when first introducing for completeness.
>
> - We used Dwork et al (2014) to give a reference for differential privacy (DP) in general when first introducing this notion. It presents a general overview of what DP is, including a full chapter (chapter 11) on machine learning.
>
> - We only cite Abadi et al. (2016)  for space limitation in the introduction. Note that we do cite Song et al. (2013) and Bassily et al. (2014) when discussing DP-SGD in Section 2.1.
>
> - The main point of this informal theorem was to show how the final error is affected by the parameters of the VN condition. One does not need to fully understand the VN condition at this point but only to see that the smaller the $\eta$, the better the convergence guarantee.
>
> - This sentence refers to the fact that the hyperparameters we tune in Section 4, especially the batch size $b$, have a much stronger impact on the final accuracy of the model when integrating DP and BR than when studying DP or BR separately. We give more detailed explanations on this in Sections 3 and 4 of the paper.
>
> - In the experiments, the overall privacy budget is fixed but not the number of iterations as we inject more or less noise according to the batch size we consider. We did not specify this in Figure 1 as it was meant to be an illustration of our experimental results but we explain this in more detail in our experimental setting (see Section 4 and Appendix D.2).
>
> **Comments on the problem setting:**
>
> - By common dataset, we mean that each worker has access to the same dataset $D$; i.e., the dataset is not partitioned across workers.
>
> - As we previously mentioned above, we agree that there exist other ways to provide DP besides Gaussian noise. However, this technique is the one that is most commonly implemented in machine learning libraries ([Opacus](https://github.com/pytorch/opacus) and [TensorFlow Privacy](https://github.com/tensorflow/privacy/tree/master/tensorflow_privacy)). When these libraries refer to DP-SGD, by default they mean Gaussian noise injection to the gradients.
>
> - **Why do we consider per-step privacy?**
> We focus on per-step privacy to show that even when sacrificing privacy (running the algorithm for an infinite time), the learning error of DP-SGD with Byzantine resilience need not vanish. Moreover, it allows us to highlight the dependence of the learning error on the model hyperparameters when injecting noise of constant variance at worker-level. This observation would not have been possible to derive if the overall privacy budget was fixed a priori. It should however be noted that in our experiments, we have been careful when computing the overall privacy budget of the learning procedure as explained in Appendix D.2.
>
> - Due to space limitation, we only provide a brief description of $(\alpha, f)$-Byzantine resilience. For further details, including intuition, we referred the reader to several prior works on this topic and provided several examples of such aggregation rules in Appendix B.

---

> > ### Author Response · Authors · 2021-11-12
> > **Response to Reviewer NXde (2/2)**
> >
> > **Comments on Section 3**
> > - We never assume the identities of Byzantine and honest workers to be known by the server. Algorithm 1 is not run on one specific entity (curator/analyst), and simply delineates the instructions run by the server and the workers. The *for* loops for the workers simply state the actions taken by them, i.e., the term "foreach" should be read as "each".
> >
> > - Theorem 1 is indeed derived from previous results from Dwork et al (2014) and Balle at al (2018), as we have pointed out above the theorem statement. Moreover, it remains  unclear to us how the given expression of $s \approx C^2\log(1/\delta)/\epsilon^2m^2$ guarantees the same level of privacy as ours. They may be close in the particular case where $(e^\epsilon - 1) m / b$ goes to $0$, but we do not think that this can be the case in general.
> >
> > - **Significance of VN condition:** The VN condition is the only verifiable condition known in the literature to guarantee $(\alpha,f)$-BR, which has allowed researchers in the past to prove Byzantine resilience of popular GARs, e.g., *Krum*, *Phocas*, *MDA*, *Trimmed Mean* and *MeaMed*, *Bulyan*. Please refer section 3.2 of our paper and to (El-Mhamdi et al., ICLR 2021) for further details.
> >
> > - This was indeed a typo, we meant to say that the privacy noise parameter $s$ is not zero, i.e., privacy is not meaningless. Correction will be made accordingly.
> >
> > - $\kappa$ indeed does not appear explicitly in Theorem 2, but that does not create a mismatch of units. $\kappa$ is actually implicitly carried within the value of $\alpha$. We did not make explicit the connection between $\kappa$ and $\alpha$ for generality, because it depends on the GAR at hand. However, the value of $\alpha$ does not change our conclusions, as pointed out in Remark 1. Furthermore, note that the paper cited by the reviewer only considers convex (and strongly-convex) settings that do not apply to most neural networks which are central to our problem. We consider a general non-convex loss function, hence we do not think that it would be fair to compare our result to these previous works.
> >
> > - Please refer to our comments above on the usefulness of per-step privacy.
> >
> > - **Contextualization of our convergence result:** The result in Corollary 1 shows how the asymptotic error (i.e., regret bound when $T$ goes to infinity) depends upon the different hyperparameters of the learning algorithm (DP-SGD with BR). This analysis allows us to re-tune the hyperparameters appropriately for improving the learning accuracy of DP-SGD in the Byzantine setting. This result is especially interesting as the asymptotic error is zero (i.e., independent of hyperparameters such as $b$) in either scenarios with only DP (i.e., no Byzantine behavior) or only Byzantine resilience (i.e., no privacy guarantees). Details on this are provided below the corollary, and also in our experiments section.

---

> > > ### Comment · Reviewer_NXde · 2021-12-01
> > > **I will leave my rating unchanged**
> > >
> > > The authors clarified some points, but I still think the paper needs work. I can't recommend acceptance in its current form.

---

### Official Review · Reviewer_RoTT · 2021-11-01

**Correctness:** 4
**Technical Novelty And Significance:** 2
**Empirical Novelty And Significance:** 2
**Recommendation:** 6
**Confidence:** 3

**Main Review:**


The paper is well-written and understandable.
The result is novel, but not very surprising.
I didn't check every detail but at a high level the results seem correct.


Detail:

In Algorithm 1, please explain the meaning (what?) and importance (why?) of "without replacement" in the sampling?  Depending on the distribution D this concept may be easier or more complicated to interpret.



**Summary Of The Paper:**


The paper studies a federated learning setting combining differential privacy (DP) and byzantine robustness (BR).  The paper shows that the "variance norm" condition for byzantine resilience needs to be relaxed as we can expect when DP is added, and then shows an adapted theorem which allows for both DP and BR.  The paper also presents a number of preliminary experiments and some interesting open questions.


**Summary Of The Review:**


The paper is mostly sound and brings a contribution (even if not very surprising).

---

> ### Author Response · Authors · 2021-11-12
> **Response to Reviewer RoTT**
>
> We thank the reviewer and provide more details on our contribution below.
>
> **On the significance of our results:** The surprising part is the degree of improvement in the cross-accuracy when increasing the batch size, especially given the fact that doing the same has very little impact when considering the Byzantine setting (with no privacy, see Appendix E) and the private setting (with no Byzantine behavior, see "No attack" curves in Figure 2) separately.
>
> **On the sampling procedure:** We consider empirical loss minimization where $D$ is a finite set of datapoints (not a generic distribution). The term "without replacement" refers to each honest worker sampling $b$ distinct datapoints from $D$ at each step. This sampling method is standard practice in privacy-preserving deep learning, and enables us to use existing results on DP.

---

### Official Review · Reviewer_tyNc · 2021-11-02

**Correctness:** 4
**Technical Novelty And Significance:** 3
**Empirical Novelty And Significance:** 3
**Recommendation:** 6
**Confidence:** 4

**Main Review:**

Strengths: This paper is well organized and easy to follow. The targeted problem is interesting and important. The theoretical contribution of this paper is providing the interplay between DP and BR.

Weakness:
1. In assumptions 1-3, it requires the empirical loss function Q to be differentiable. Is that possible to extend current results for non-smooth loss? Say, the absolute derivation loss.
2. As it has been claimed in the abstract, "by carefully re-tunning the learning algorithm", how to choose the learning rate $\gamma_1,\dots,\gamma_T$? Is the proposed method robust with a certain range of learning rates?

**Summary Of The Paper:**

This paper combined differential privacy and byzantine resilience in the distributed SGD algorithm.  The authors provide a simultaneous theoretical guarantee of DP and BR by re-tuning the algorithm. Both theoretical results and numerical experiments are conducted to show the effectiveness.

**Summary Of The Review:**

1. In assumptions 1-3, it requires the empirical loss function Q to be differentiable. Is that possible to extend current results for non-smooth loss? Say, the absolute derivation loss.
2. As it has been claimed in the abstract, "by carefully re-tunning the learning algorithm", how to choose the learning rate $\gamma_1,\dots,\gamma_T$? Is the proposed method robust with a certain range of learning rates?

---

> ### Author Response · Authors · 2021-11-12
> **Response to Reviewer tyNc**
>
> We thank the reviewer and provide some additional feedback below.
>
> **Answer to Q1:** We believe that the assumptions we make are fairly classical in optimization problems, hence we did not investigate non differentiable functions at the moment. We however believe our analysis can be adapted to non differentiable functions and this would be an interesting future direction for our work.
>
> **Answer to Q2:** In our paper, we only tune the hyperparameters appearing in the asymptotic error (first input to the max function in Corollary 1), which is independent of the learning rate as shown in Corollary 1. In practice, we choose a constant learning rate so that the cross-accuracy reaches the maximum level after 300 steps (in the non-private Byzantine-free setting). Our experimental findings (especially on the impact of hyperparameter tuning) are actually consistent across different learning rates.

---

### Official Review · Reviewer_Vukg · 2021-11-08

**Correctness:** 3
**Technical Novelty And Significance:** 1
**Empirical Novelty And Significance:** 2
**Recommendation:** 3
**Confidence:** 3

**Main Review:**

The paper is clearly written and provides a reasonable solution to this very interesting question. However, I do have several comments.
1. I wonder if there is any difference between the Byzantine model in the paper and the adversarial contamination model in the robust statistics community (see papers cited below). In my opinion they are the same as in both models a fraction of data are allowed to be arbitrarily changed. In particular, the interesting part of Byzantine seems to be lost since the aggregator is always trust-worthy.
2. If they are the same, then I would like to see a modular solution based on, for example, the following
    > Aggregating the gradient accurately implies SGD convergence

    Starting from here, features like robustness(=BR assuming the answer to my Q1) and/or DP can be added to the aggregator. There are robust and accurate aggregators with much nicer theoretical guarantee, e.g. [this paper](https://epubs.siam.org/doi/abs/10.1137/17M1126680). There's actually a big line of work.
    There is also [accurate robust + DP aggregators](https://arxiv.org/abs/2102.09159).
    Based on what is presented in the paper, I don't see a necessary interplay between robustness=BR and SGD convergence, so I think a modular solution is possible and better, potentially encouraging deeper observation for practice.
    In particular, the aggregators used in the paper seem to be variants of geometric medians and other versions of medians, which is one of the central objects of robust statistics. The adaptation seems to be rooted in the VN condition, which doesn't seem "widely used" as claimed in the paper. Almost all references of it are written by roughly the same group of authors. The notion of $(\alpha,f)$-BR also seems to be restricted to the same group. Following this line of work may not be the best approach and I highly recommend the authors to explore the connections and make use of existing powerful results, rather than re-inventing sub-optimal wheels.
    Of course all these comments rely on the answer to my Q1, which I might have misunderstood. I would love to hear the reply and change my rating correspondingly.
3. It's unclear to me why RHS of the bound in Corollary 1 is decreasing in $b$. The relevant part in this bound looks like $\ln(b)(C+1/b)^2$. I tested with a few different $C$ and some make it increasing while others make it decreasing. It is important to justify this point since it is one of the most important practical implications of the theory in this paper.

**Summary Of The Paper:**

The paper considers the natural class of algorithms, namely *Aggregators with Gaussian noise* for distributed SGD with differential privacy (DP) and Byzantine resilience (BR). Previous results shows VN $\Rightarrow$ BR $\Rightarrow$ convergence of SGD.
The authors first show that aggregators with Gaussian noise algorithms satisfy DP but violates VN necessarily, so approximate VN is proposed. Theorem 2 shows approximate VN $\Rightarrow$ convergence. Proposition 2 shows the above algorithms satisfies approximate VN with certain parameters. With the combined bound Corollary 1, the authors observe (and then verify by experiments) that larger batch size is beneficial and in particular more beneficial than when DP or BR is enforced alone.


**Summary Of The Review:**

Not recommended as its current form since it seems to have missed a deeper connection and hence a much nicer solution. The batch size observation also doesn't seem to be firmly supported. However, the question is interesting so I look forward to its future form.

---

> ### Author Response · Authors · 2021-11-12
> **Response to Reviewer Vukg**
>
> We thank the reviewer for the feedback and provide our answers to their comments below.
>
> **Answer to Q1:** Note that in our case, the server is not trusted, which is the main reason why our problem is challenging from a robust estimation perspective. We clarify this statement below by comparing the trusted server and non-trusted server settings.
>
> - If the server were trusted, it would have direct access to the workers' data and would only apply confidentiality when computing the robust statistic. In this case, as the reviewer points out, Byzantine resilience and robust estimation share some similarities. However, note that while the two problems seem close, in general, optimizing a non-convex function from corrupted gradients is more difficult than simply estimating the mean of a distribution from corrupted data samples.
>
> - In our context, the server is not trusted but rather honest-but-curious (see the threat model in Section 2). Therefore, the privacy mechanism is applied at the worker level before sending the gradients to the server for computing the robust statistic. In this case, the references provided by the reviewer are no longer applicable. The honest-but-curious assumption indeed makes the computation of robust statistics much more challenging, especially in non-convex settings where workers only send gradients.
>
> **Answer to Q3:** We agree with the reviewer on this point. The function $b \mapsto \ln(b) (C + 1/b)^2$ need not be decreasing on $[1,m]$, especially when $C$ is not sufficiently small. However, by analyzing the variation of the function, we can see  that it is indeed decreasing on $(1,m]$ as soon as $C < \frac{2\ln(m) - 1}{m}$, which is equal to $3.5 \times 10^{-4}$ in our experiments ($m = 60,000$ for MNIST and Fashion-MNIST). In our paper, $C = \frac{1}{m(e^\epsilon - 1)}$, which is practically small. For example, in our experiments, $C$ is always smaller than $3.25 \times 10^{-4}$. Hence, the function is decreasing on the entire range of $b$. This observation will be consistent for many learning problems as soon as the size of the dataset $m$ is sufficiently large. Appropriate remarks will be made in the paper.

---

> > ### Comment · Reviewer_Vukg · 2021-11-13
> > **reply**
> >
> > ### Regarding Q1 and Q2
> > Let me point out what is confusing me (and potentially other reviewers and future readers) here. Even when no workers is Byzantine, the model is still novel/non-standard and requires some explanation which is missing in the current paper. More specifically, the algorithm looks like
> > 1. Compute $n$ mini-batch gradients at the same parameter $\theta_t$ in parallel. Each worker is in charge of one gradient.
> > 2. Adversary collects all $n$ gradients and update $\theta_{t+1}$ and send to all workers.
> >
> > Because the adversary sees all gradients, the entire mechanism actually involves two levels of DP composition: 1. non-adaptive composition over workers 2. adaptive composition over iterations. It makes sense to first show per-step and per-worker privacy, but then you need to make the above argument clear and shouldn't have left it to the readers.
> >
> > In addition, it is similar to but different from local DP since all workers have access to the entire dataset. This should also be explicitly noted as it has already misled me and reviewer NXde.
> >
> > Because of this special structure, the robust DP mean estimation paper I mentioned no longer applies. You are right that "optimizing a non-convex function from corrupted gradients is more difficult than simply estimating the mean of a distribution from corrupted data samples", but it doesn't prevent robust estimation from being a subroutine for SGD. I still have serious concern that all these aggregators are re-inventing suboptimal wheels. In particular, B.3 introduces coordinate median whose breakdown point is 1/2, which seems to be equivalent to $n\geqslant 2f+1$. Despite the connection, none of these aggregator papers cited any robust statistics papers, so it is possible that the two communities are unaware of each other. I highly recommend checking out the literature before re-submission.
> >
> > I said "the interesting part of Byzantine seems to be lost since the aggregator is always trust-worthy". Here trust-worthy means it is always assumed to follow the protocol, not that it is not curious about private information. The interesting part of Byzantine is still lost. Wording like this leaves no room for a genuinely Byzantine paper, which is unfair in my opinion.
> >
> > ### Regarding Q3
> > I think you need an additional claim in Corollary 1. This is one of your main message, so it must be clear. You should think of other ways to polish the presentation. All these extra conditions you gave me in the reply make your message more limited and less convincing.

---

> > > ### Author Response · Authors · 2021-11-22
> > > **Re: reply**
> > >
> > > **Setup:** Thank you for pointing this out, the setup you presented is exactly the one we consider. We will make it more explicit in the next version of the paper. We will also add appropriate discussions on the difference with exiting settings.
> > >
> > > **Mean estimation:** Although the robust mean estimation problem shares some similarities, their applicability to non-convex learning is limited from the fact that the distribution of workers’ gradients need not satisfy the assumptions made in robust mean estimation works, e.g., gradients need not be Gaussian or even sub-exponential. Nevertheless, we will include the relevant discussion on this point.
> > >
> > > **Trustworthy server:** It still is unclear to us how the interesting part is lost when the aggregator is assume to correctly follow the protocol. Note that if the server is also assumed Byzantine, the problem is rendered meaningless and unsolvable. Indeed, in the parameter-server architecture, the workers only communicate with the server and inter-worker communication is absent. The server is also the only entity that is allowed to update the model. Hence a Byzantine server can trivially prevent the procedure from learning anything.
> > >
> > > Note also that, even when the server is assumed to follow the protocol correctly, the problem is not trivial. In fact, as the identity of the Byzantine workers is a priori unknown and they could be omniscient, the whole difficulty lies in the computation of a resilient aggregation (without assuming anything about the distribution of the honest gradients).
> > >
> > > **Extra conditions:** We thank again the reviewer for pointing out this element about our claim. We will indeed add a clear discussion on this point as an additional claim in the next version of the paper. However, as we explained in our response above, we do not believe this to be a critical limitation of our work as the extra condition will be met in practice.

---

### Decision · Program_Chairs · 2022-01-20

**Decision:**

Reject

**Comment:**

The paper considers the natural class of algorithms, namely Aggregators with Gaussian noise for distributed SGD with differential privacy (DP) and Byzantine resilience (BR). Previous results shows VN->BR-> convergence of SGD. The authors first show that aggregators with Gaussian noise algorithms satisfy DP but violates VN necessarily, so approximate VN is proposed. Theorem 2 shows approximate VN->convergence. Proposition 2 shows the above algorithms satisfies approximate VN with certain parameters. With the combined bound Corollary 1, the authors observe (and then verify by experiments) that larger batch size is beneficial and in particular more beneficial than when DP or BR is enforced alone. In the formulation, an important baseline of robust mean aggregation [Diakonikolas,Kamath,Kane,Li,Moitra,,Stewart'2016] and even more relevant baseline of robust and DP mean aggregation[Liu,kong,Kakade,Oh,'21] are somehow missing. One would assume that directly applying these well-known techniques might give the desired DP and robust SGD. The field at the intersection of differential privacy and robustness has evolved quite a bit recently and tremendous technical innovations are happening. Given the relveance of the proposed problem to this line of work, one should make the connections precise and explain the differences.

---

> ### Public Comment · ~Nirupam_Gupta1 · 2022-02-01
> **The problem considered in [Liu,kong,Kakade,Oh,'21] is not the same as in our paper**
>
> We thank the reviewers and the PC for providing valuable comments on our work.
>
> To clarify the problem statement to the community, we believe it is important to highlight the critical difference between the problem setup considered in our paper and the one in robust and DP mean aggregation[Liu,kong,Kakade,Oh,'21]. In the latter, the server is assumed trusted and is the one responsible for ensuring DP of the workers' data. However, in our problem setup, the server is NOT trusted and each worker is responsible for ensuring DP of its data at its own end. This is a crucial difference, and indeed what we show in our paper is that DP and Byzantine resilience approaches are at odds in our setup, unlike the setup considered in [Liu,kong,Kakade,Oh,'21]. Moreover, we believe that their approach cannot be trivially adapted to solve our problem. We will make this difference clear in our next submission.